# FAM46C/TENT5C functions as a tumor suppressor through inhibition of Plk4 activity

Karineh Kazazian[1,2], Yosr Haffani[3], Deanna Ng[1], Chae Min Michelle Lee[1], Wendy Johnston[4], Minji Kim[1], Roland Xu[1], Karina Pacholzyk[1], Francis Si-Wah Zih[2], Julie Tan[1], Alannah Smrke[1], Aaron Pollett[5], Hannah Sun-Tsi Wu[5] & Carol Jane Swallow[1,2 ✉]

Polo like kinase 4 (Plk4) is a tightly regulated serine threonine kinase that governs centriole duplication. Increased Plk4 expression, which is a feature of many common human cancers, causes centriole overduplication, mitotic irregularities, and chromosomal instability. Plk4 can also promote cancer invasion and metastasis through regulation of the actin cytoskeleton. Herein we demonstrate physical interaction of Plk4 with FAM46C/TENT5C, a conserved protein of unknown function until recently. FAM46C localizes to centrioles, inhibits Plk4 kinase activity, and suppresses Plk4-induced centriole duplication. Interference with Plk4 function by FAM46C was independent of the latter's nucleotidyl transferase activity. In addition, FAM46C restrained cancer cell invasion and suppressed MDA MB-435 cancer growth in a xenograft model, opposing the effect of Plk4. We demonstrate loss of FAM46C in patient-derived colorectal cancer tumor tissue that becomes more profound with advanced clinical stage. These results implicate FAM46C as a tumor suppressor that acts by inhibiting Plk4 activity.

[1] Lunenfeld Tanenbaum Research Institute, Sinai Health System, Toronto, ON M5G 1X5, Canada. [2] Department of Surgical Oncology, University of Toronto, Toronto, ON M5G 2M9, Canada. [3] Laboratory of Physiopathology, Alimentation and Biomolecules LR17ES03, Higher Institute of Biotechnology, Sidi Thabet, University of Manouba, Ariana 2020, Tunisia. [4] Department of Radiation Oncology, University of Toronto, Toronto, ON M5T 1P5, Canada. [5] Department of Laboratory Medicine and Pathology, University of Toronto, Toronto, ON M5S 1A8, Canada. ✉email: carol.swallow@sinaihealth.ca

Polo-like kinase 4 (Plk4) is a serine threonine kinase found most abundantly at the centriole[1–3], but also localized to the MTOCs of acentriolar cells[4] and to the lamellipodia of migrating cells[5]. Plk4 is required for centriole duplication, and its activity is closely tied to cell cycle progression, increasing during S phase and peaking at G2/M[1,6,7]. Increased Plk4 expression is found in a spectrum of common human malignancies[8–11], and results in centriole overduplication and aneuploidy[12–16]. While germline Plk4 haploidy is insufficient to suppress tumor development in adult life[17], inhibition of Plk4 kinase activity reduces cancer cell proliferation and migration[5,9,18], indicating a potential for oncogenic function. In keeping with the latter, Plk4 depletion from malignant cells inhibits tumor progression in xenograft models[9,19], while upregulation of Plk4 can promote tumorigenesis in vivo[13,16,20]. With this preclinical evidence as well as its association with aggressive tumor behavior and chemo-resistance in breast cancer patients, Plk4 is currently under investigation as a therapeutic target in Phase I/II clinical trials[21]. Identification of an expanded Plk4 bio-interactome has sparked further interest in the therapeutic potential of its members[19].

Plk4 is a pivotal actor in the precise and highly regulated process of centriolar duplication, a process that is evolutionarily conserved in eukaryotes. The effectors that act downstream of Plk4 to realize spatially correct recruitment of microtubules to the nascent daughter centriole include STIL, SAS-6, and CEP135/CPAP[7,22–28]. Upstream regulation of Plk4 itself remains incompletely understood, but appears to involve multiple sophisticated systems to maintain tight temporal and spatial control of both protein level and kinase activity. Plk4 is localized to the centriole through interactions with scaffolding proteins[29–31], and its kinase activity is subject to regulation by autophosphorylation[24,32–35], acetylation[36] and dephosphorylation[37,38]. Plk4 substrate STIL, together with its interactor CEP85, in turn regulate Plk4 kinase activity and localization, critically contributing to the control of centriolar duplication[7,23,24,39,40]. Plk4 protein is rapidly degraded through βTRCP-mediated ubiquitination, which is triggered by Plk4 trans-autophosphorylation within a phosphodegron in the linker region positioned between the kinase domain and Polo Box domains[33,35,41]. Intramolecular interaction between the third polo box motif (PB3) and the linker serves to suppress kinase activity, and disruption of this interaction, for instance through binding to STIL[28], relieves the inhibition, culminating in phosphorylation of Plk4 substrates[34]. Tight regulation of Plk4 kinase activity is essential to maintain precise coordination of the centrosome and cell cycles, ensuring diploidy and avoiding other adverse consequences of centriolar overduplication[12,13,16,32].

In a search for additional functional Plk4 interactors, we identified FAM46C/TENT5C, a ≈ 47 kDa protein recently confirmed to function as a non-canonical poly(A) RNA polymerase in multiple myeloma cell lines[42]. Here we show that FAM46C localizes to centrioles throughout the cell cycle, physically interacts with Plk4 kinase/PB-1/PB-2 domains, and impairs Plk4 kinase activity, restraining centriole duplication. In a spheroid model, FAM46C depletion promoted invasion of HeLa cancer cells into the surrounding matrix. When Plk4 was inactivated by centrinone B, however, FAM46C depletion had no effect, indicating that the suppressive effect of FAM46C on cancer cell invasion is mediated through its inhibition of Plk4 activity. Furthermore, in a mouse xenograft model of human cancer using MDA MB-435 cells, FAM46C inhibited tumor progression in opposition to Plk4. In the cancer cell lines studied here, the inhibitory effect of FAM46C on Plk4 activity was not dependent on its poly(A) RNA polymerase function. We also show for the first time that unlike other centriolar Plk4 interactors, FAM46C is depleted in human colorectal cancer tumor tissue, compared with paired normal mucosa samples taken from the same patient.

Furthermore, the FAM46C/Plk4 ratio declined markedly with advancing clinical cancer stage. Taken together, these results reveal a tumor suppressor function of FAM46C/TENT5C through inhibition of Plk4 kinase activity.

## Results

**FAM46C interacts with Plk4 and localizes to the centriole**. Of the sixty-five and seventeen potential Plk4 interactors identified by yeast 2– hybrid (Y2H) screens of the *Drosophila* and human ORFeomes, respectively (DroID:, http://www.droidb.org and HuRI: http://interactome.baderlab.org), only four were shared between species (Fig. 1a): Plk4 itself, an interaction that is expected given its known functional homodimerization[34,35,43]; βTRCP, also a well-known physical and functional interactor across species[33,35,41,44]; FAM46C; and FAM46B. The FAM46/TENT5 proteins are conserved across eukaryotes (Supplementary Fig. 1a, b)[45] but were until recently of unknown function. Human FAM46C and FAM46B are two of four differentially expressed family members (Supplementary Fig. 1a, c), that based on sequence were predicted to function as non-canonical poly(A) RNA polymerases[45], with recent confirmation of selective mRNA stabilization by FAM46C in multiple myeloma cells[42]. While FAM46A and FAM46D interacted with several other proteins in Y2H screens (Supplementary Fig. 1d;[46,47]), they did not interact with Plk4. Of the four HsFAM46/TENT5 family members, only hFAM46C interacted with hPlk4 in reciprocal co-immunoprecipitation experiments (Fig. 1b).

An antibody raised to full-length hFAM46C stained centrioles, confirmed by centrin positivity (Fig. 1c, d). At high power, the centriolar localization of endogenous FAM46C partially overlapped with that of endogenous Plk4 (Fig. 1c). Upon siFAM46C-mediated depletion, endogenous FAM46C was no longer visualized at centrioles (Fig. 1d). Interestingly, a rosette-type centriolar over-duplication phenotype was frequently observed in FAM46C-depleted cells. RFP-tagged hFAM46C also localized specifically to centrioles, partially overlapping with endogenous Plk4 (Fig. 1e), and with the other centriolar proteins CPAP, CP110, and CEP135 (Supplementary Fig. 2a). Of note, centriolar staining for FAM46C, whether endogenous or exogenous, was consistently observed at centrin-positive structures in all cells, and FAM46C appeared to be a ubiquitous component of centrioles. In reciprocal co-immunoprecipitation experiments, FAM46C did not interact with CEP135, nor with centriolar scaffolding proteins CEP152 or CEP192 (Supplementary Fig. 2b–d). RFP-FAM46C was found in the same centriolar fraction as FLAG-Plk4 in extracts from doubly transfected cells that were separated by sucrose gradient (Supplementary Fig. 3a). Taken together, these data suggested the possibility of a specific functional interaction between FAM46C and Plk4.

**FAM46C regulates centriole duplication in a Plk4-dependent manner**. To investigate the function of FAM46C, knockdown was achieved by shFAM46C (four individual constructs), confirmed by RT-PCR and immunoblot (Fig. 2a). FAM46C depletion led to an increase in centriolar number in U2OS cells, with an increased proportion of cells that had >4 centrioles, and a corresponding decrease in cells with 2–4 centrioles (Fig. 2b), again frequently accompanied by development of a centriolar phenotype that was reminiscent of the well-described Plk4 overexpression rosette (Fig. 2b[25,48]). The supernumerary centrioles appeared structurally normal by electron microscopy, which also revealed evidence of multiple daughter centrioles arising from the same mother, as typically seen with Plk4 overexpression (Supplementary Fig. 3b). The supernumerary centrioles generated in response to FAM46C depletion also each stained positive for SAS-6, in keeping with

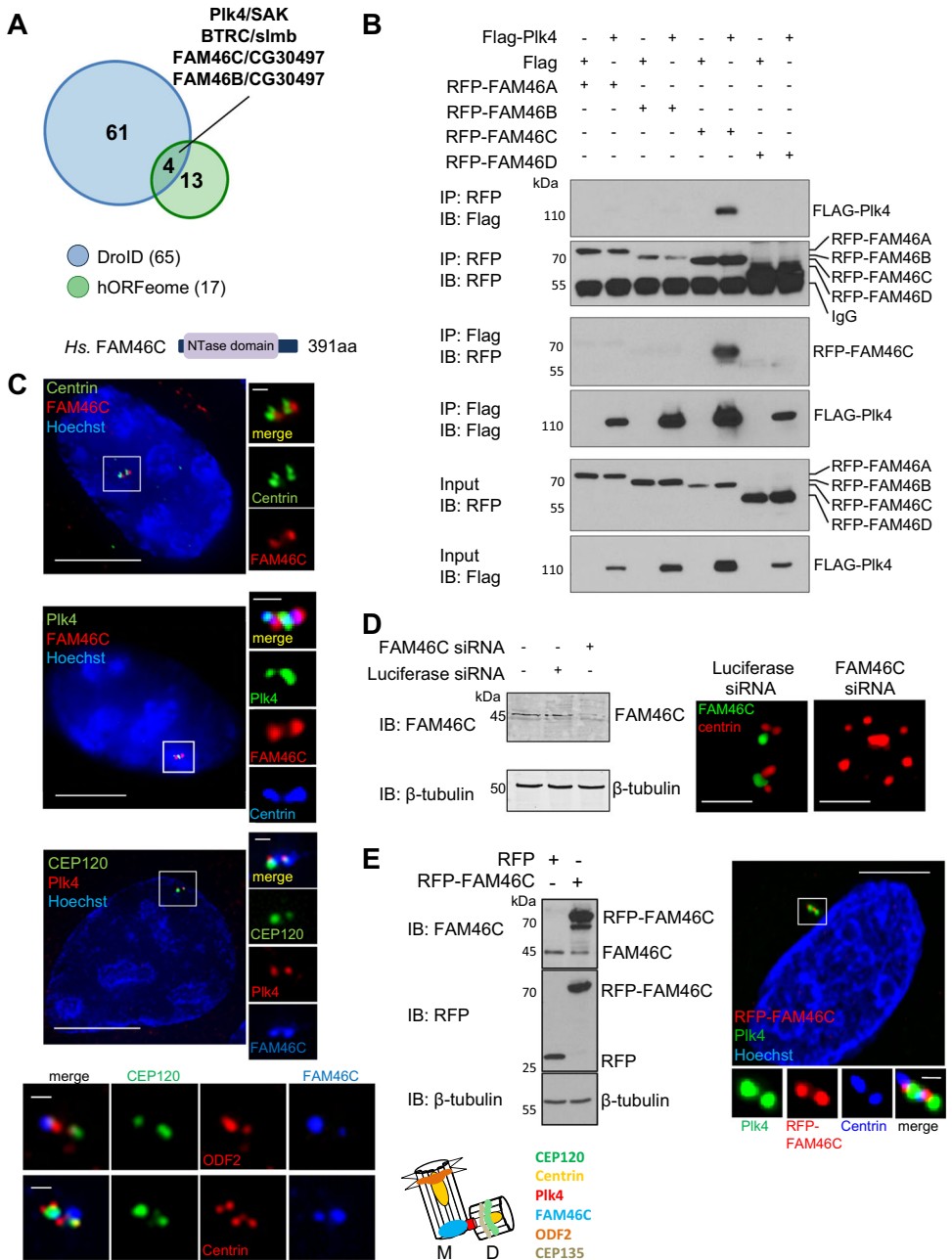

their generation via the Plk4 pathway (Fig. 2c;[7,24,26]). Further-more, these SAS-6-positive and centrin-positive foci were arranged in a typical Plk4 overexpression rosette.

Of note, the centriolar over-duplication phenotype observed with FAM46C depletion was lost after ≈3 weeks' culture of shFAM46C cells, in association with recovery of FAM46C expression, which could potentially reflect a proliferative disadvantage[12,14,15]. Given that in myeloma cell lines, FAM46C's RNA polymerase activity regulates transcription of a suite of genes that control proliferation and arrest[42], we investigated the acute effect of FAM46C depletion on viability and growth in culture. At 48 h in culture, MDA MB-435 cells depleted of FAM46C showed no significant difference in viability or proliferative rate vs. controls, while simultaneously displaying a higher number of centrioles/cell (Fig. 2d). Thus, it appeared unlikely that cell cycle arrest or delay accounted for the increase in centrioles/cell. It is noteworthy that the centriole

overduplication phenotype induced by FAM46C depletion was observed not only in U2OS cells, which typically tolerate and display a high level of centriole amplification, but also in the melanoma line MDA MB-435.

In HeLa cells synchronized by Aphidicolin block and release, FAM46C staining was observed at centrioles throughout the ensuing phases of the cell cycle (Supplementary Fig. 4a, b). Co-staining with antibody to ODF2, which localizes preferentially to the mother centriole, or with antibody to CEP120, which localizes preferentially to the daughter centriole, showed that FAM46C is localized predominantly to the mother centriole (Fig. 1c, bottom panels; Supplementary Figs. 4b, c; 5a). Super resolution fluorescence imaging of U2OS cells confirmed FAM46C localization to centrioles, and revealed a ring-like distribution on the mother centriole, adjacent to and partially overlapping with Plk4 (Supplementary Fig. 4d). In T-REx YFP-Plk4 U2OS cells induced to display a Plk4-driven centriole overduplication phenotype by

**Fig. 1 FAM46C is a conserved protein that interacts with Plk4. a** Diagram summarizing results of yeast two-hybrid screens for Plk4/SAK interactors from the HuRI (the human reference protein interactome mapping project, http://interactome.baderlab.org) and DroID (Drosophila interactions database, http://www.droidb.org) databases, showing overlap of four interactors, including *H. Sapiens* FAM46C/*D. Melanogaster* CG30497. *Hs.*FAM46C is a protein of 391 amino acids (aa) that contains a nucleotidyltransferase (NTase) domain[45], that has been recently renamed TENT5C. **b** Immunoblots of Flag-Plk4 and four individual *Hs.*RFP-FAM46 proteins after coexpression in HEK293T cells, showing reciprocal co-immunoprecipitation of Plk4 only with FAM46C. Uncropped blots for this and subsequent figures can be found in Supplementary Fig. 10. **c** Localization of FAM46C to centrioles, identified by staining with centriole markers including centrin, CEP120 (daughter centriole marker) and ODF2 (mother centriole marker), and overlap with Plk4. Representative immunofluorescence images of U2OS cells labeled with antibodies to: top panel, centrin (green) and FAM46C (red); second panel, Plk4 (green), FAM46C (red), centrin (blue); third panel, CEP120 (green, daughter), Plk4 (red), FAM46C (blue), and with Hoechst (blue). The right panels/inserts show magnified centrosomes (boxed in white). Bottom panels show magnified centrosomes labeled with antibodies to CEP120 (green, preferential labeling of daughter), ODF2 (red, top row, preferential labeling of mother), centrin (red, bottom row) and blue (FAM46C). FAM46C localizes predominantly to the mother centriole in unsynchronized cells (see Supplementary Fig. 4b for localization in synchronized cells). Cartoon summarizes putative FAM46C localization in relation to centriolar markers probed in this study, showing localization at the proximal mother centriole (M mother, D daughter; Bar 1 μm). **d** Left panel, representative immunoblot of Luciferase or FAM46C siRNA transfected U2OS cell extracts probed using anti-FAM46C antibody, with β-tubulin as loading control, demonstrating 87% depletion of FAM46C. Right panels, representative immunofluorescence images of Luciferase or FAM46C siRNA-transfected U2OS cell centrosomes labeled with antibodies to centrin (red) and FAM46C (green). FAM46C depletion was confirmed and resulted in a rosette-like centriolar phenotype. Bar: 1 μm. **e** Left panel, immunoblot showing expression of RFP and RFP-FAM46C in U2OS cells transfected with RFP or RFP-FAM46C, respectively, using anti-FAM46C and anti-RFP antibodies with β-tubulin as a loading control, representative of $n = 3$ independent experiments. Right panel, localization of exogenous FAM46C to centrioles shown in representative immunofluorescence images of U2OS cells transfected with RFP-FAM46C (red) for 42 h and labeled with antibodies to Plk4 (green) and centrin (blue, bottom panels/inserts) and with Hoechst (blue, top panel). The bottom panels/inserts show a magnified centrosome (boxed in white). Bars: 10 μm, inserts 1 μm.

treatment with tetracycline, FAM46C appeared to be concentrated at the central mother centriole, surrounded by CEP120- and Plk4- positive nascent daughters (Supplementary Fig. 4c, d).

Depletion or inhibition of Plk4 results in gradual loss of centrioles over subsequent cell cycles; in a variety of human cell lines, treatment with siPlk4 for 48–72 h results in a significant increase in the proportion of cells with only 1 centriole[2,3]. In U2OS cells depleted of Plk4, some FAM46C staining was still observed on the single residual centriole (Supplementary Fig. 5a). Increased FAM46C expression was also associated with an altered centriolar profile: a higher proportion of RFP-FAM46C cells had only 1 centriole, while fewer possessed >4 centrioles, as compared with RFP control (Fig. 2e, Supplementary Fig. 5b). The shFAM46C centriole phenotype was partially rescued by RFP-FAM46C (Supplementary Fig. 5b), arguing against an off-target effect and providing further evidence that FAM46C was indeed regulating centriole number.

The localization of FAM46C and Plk4 to centrioles throughout the cell cycle (Supplementary Fig. 4b;[1,3]), in addition to the physical interaction of FAM46C with Plk4 observed in co-immunoprecipitation experiments, taken together with the rosette-like centriolar configuration conferred by depletion of FAM46C, suggested that FAM46C might normally function to limit Plk4 activity and/or abundance. In addition, expression of FAM46C was cell cycle-dependent and peaked in mitosis (Supplementary Fig. 6), similar to that of Plk4[1,49]. Indeed, Plk4-induced centriole overduplication was abrogated by RFP-FAM46C, manifest as both reduction in centriole number and suppression of rosette formation (Fig. 3a). Furthermore, the centriolar overduplication phenotype induced by depletion of FAM46C was not observed when Plk4 was itself depleted (Fig. 3b). While it is conceivable that epistasis could account for the inability of FAM46C depletion to overcome the centriolar phenotype induced by Plk4 depletion, a more likely explanation is that the centriolar overduplication observed upon FAM46C depletion is dependent on Plk4 expression.

In cells treated with hydroxyurea to effect an S phase arrest, the centriole cycle becomes uncoupled from the cell cycle, and centriole duplication continues, with a resultant increase in centriole number; this phenomenon is augmented by tetracycline-induced Plk4 expression (Supplementary Fig. 5c). In cells under hydroxyurea arrest, which would minimize any potential differences in cell cycle progression, FAM46C overexpression nevertheless prevented the increase in centriole count per cell otherwise seen with Plk4 upregulation (Supplementary Fig. 5d), showing that this effect was intimately linked to the centriole cycle, and supporting the hypothesis that FAM46C acted in opposition to Plk4.

**FAM46C inhibits Plk4 autophosphorylation and interacts with the Plk4 kinase and PB-1/PB-2 domains.** We surmised that FAM46C inhibited Plk4 function, and examined its effect on Plk4 kinase activity. In an in vitro kinase assay, the autophosphorylation of full-length wild-type FLAG-Plk4 was markedly reduced in the presence of FAM46C, with a clear dose response (Fig. 4a). By contrast, comparable levels of FAM46A had no discernible effect on Plk4 autophosphorylation (Supplementary Fig. 7a). While the association between Plk4 and FAM46C demonstrated in Y2H and reciprocal co-immunoprecipitation experiments (Fig. 1b) suggested a direct interaction, it was possible that FAM46C instead interfered with activation of Plk4 by upstream regulators, in particular STIL and/or CEP85[24,39]. To explore this, we performed an in vitro kinase assay using His-tagged Plk4 kinase domain+linker (AA1–390) purified from bacterial extract on a nickel column. By this method, we were able to generate and purify large amounts of Plk4(1–390) and GST-FAM46C and test their interaction independent of other cellular proteins. Autophosphorylation of the Plk4 fragment was inhibited by purified GST-FAM46C, with a dose-response effect (Fig. 4b), indicating that the inhibitory effect of FAM46C on Plk4 kinase activity was mediated through a direct physical interaction between the two proteins. In transient co-expression experiments, increasing levels of FAM46C appeared to stabilize wild-type Plk4 protein (Fig. 4c). This would be consistent with reduced Plk4 autophosphorylation, required for its ubiquitination and degradation, providing further evidence for inhibition of Plk4 kinase activity by FAM46C. Centriolar loading of SAS-6 is another read-out of Plk4 activity, since it is dependent on phosphorylation of its interacting partner STIL by Plk4[7,24,26]. As noted, upon FAM46C depletion, SAS-6 was observed at each of the nascent centrioles in the resulting centriolar rosette (Fig. 2c), confirming the downstream effect of upregulated Plk4 activity, of which this centriolar configuration is a hallmark.

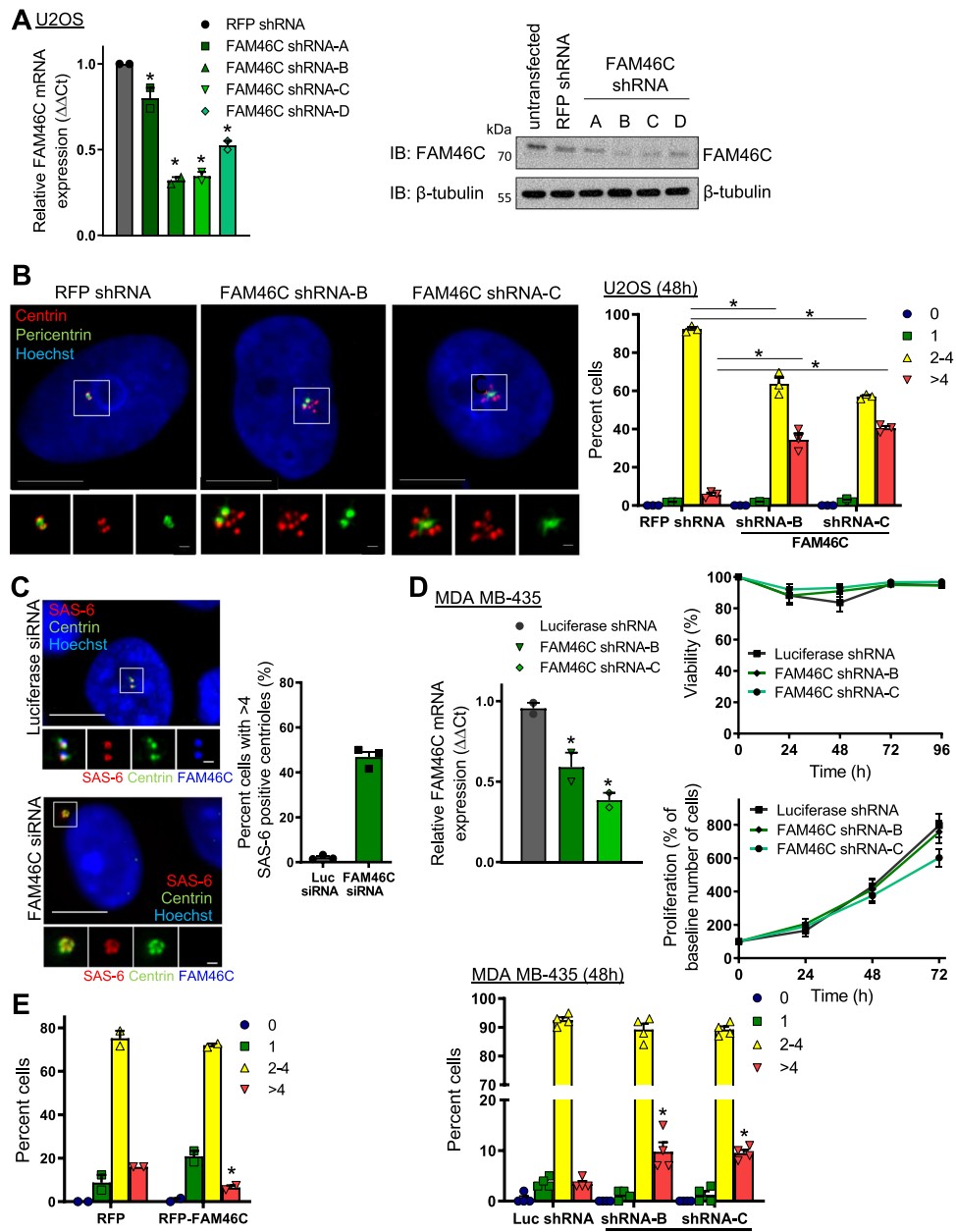

**Fig. 2 FAM46C regulates centriole duplication. a** Depletion of FAM46C in U2OS cells using four individual FAM46C shRNAs, confirmed by reduction of FAM46C mRNA levels (left panel) and reduction of FAM46C protein levels (right panel) relative to RFP shRNA cell lines, *n* = 2 independent experiments, *p < 0.001 vs. RFP shRNA. **b** Centriolar overduplication phenotype in U2OS cells depleted of FAM46C. Left panels show representative immunofluorescence images of U2OS cells labeled with antibodies to centrin (red) and pericentrin (green), and with Hoechst (blue). The bottom panels/ inserts show magnified centrosomes (boxed in white) for each condition. Right panel, bar graph showing proportion of cells with indicated number of centrioles per cell, quantified by scoring centrin-positive foci. *n* = 3 independent experiments with >50 cells measured in each, *p < 0.001 vs. RFP shRNA. **c** Downstream marker of Plk-4 driven centriolar duplication SAS-6 is amplified by FAM46C depletion, as shown in representative immunofluorescence images (left panels) of Luciferase- or FAM46C- siRNA transfected U2OS cells labeled with antibodies to SAS-6 (red), centrin (green), FAM46C (blue, bottom panels/inserts) and with Hoechst (blue, top panels). Right panel: Bar graph showing proportion of cells with more than four SAS-6 positive centrioles, *n* = 3. Bars: 10 μm; insert 1 μm. **d** Depletion of FAM46C in MDA MB-435 cells using two individual FAM46C shRNAs. Left panel, confirmation of reduction of FAM46C mRNA levels relative to Luciferase shRNA cell lines, *n* = 2 independent experiments, *p = 0.016 vs. Luciferase shRNA. Right panels, viability and proliferation of MDA MB-435 cells, treated as indicated. Cells were labeled with Hoechst and Propidium Iodide, then imaged using the Celigo Cell Imaging Cytometer at the indicated times. Dead cells were distinguished from the live cells based on the mean intensity of the Propidium Iodide signal. Number of independent experiments was 3 for Viability, 3 for Proliferation. p = NS vs. control, using ANOVA with Bonferroni correction. Bottom panel, bar graph showing proportion of cells with indicated number of centrioles per cell, quantified by scoring centrin-positive foci. *n* = 4 independent experiments with >60 cells measured in each, *p < 0.015 vs. Luciferase shRNA. **e** Reduction in centriole number in response to RFP-FAM46C expression for 42 h in U2OS cells. Bar graph shows proportion of cells with indicated number of centrioles per cell, quantified by scoring centrin-positive foci. *n* = 2 independent experiments with >50 cells measured in each, *p = 0.0098 vs. RFP. Bars: 10 μm, inset 1 μm. Data are means ± SEM.

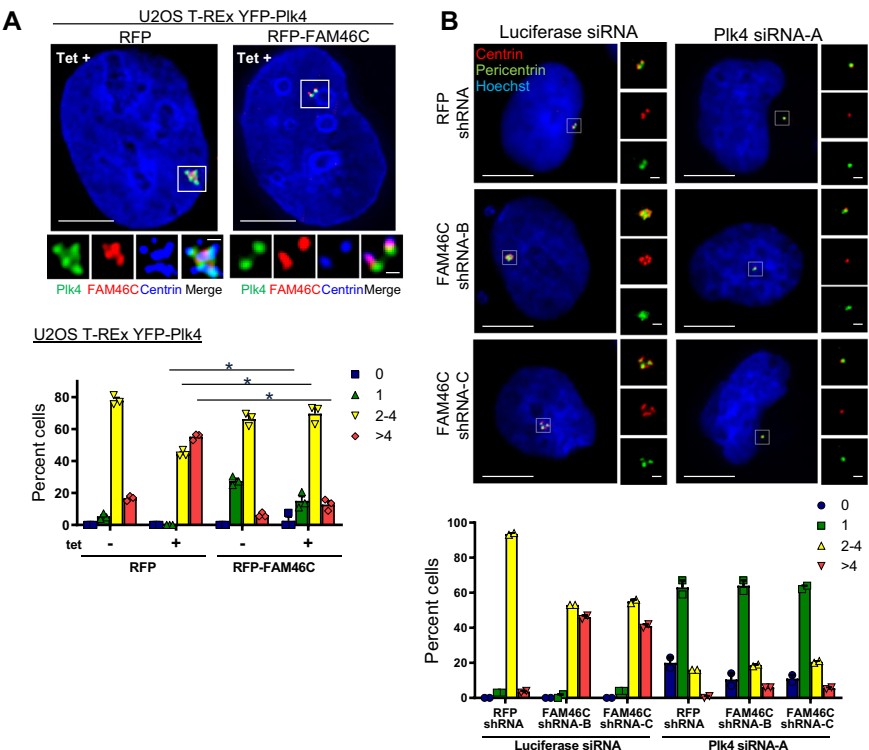

**Fig. 3 FAM46C interacts functionally with Plk4 to regulate centriole duplication. a** Suppression of Plk4 overduplication phenotype by FAM46C shown in representative immunofluorescence images of U2OS T-REx YFP-Plk4 (green) cells with Plk4 expression induced by tetracycline (Tet+) and transfected with RFP or RFP-FAM46C X40h (top panels), with quantification of centriole numbers (bottom panel). Cells were labeled with antibodies to FAM46C (red, stains endogenous FAM46C in left panel, and both endogenous and transfected FAM46C in right panel), and centrin (blue, inserts), and with Hoechst (blue, top panels). The inserts show magnified centrosomes (boxed in white) for each condition. Bar graph shows proportion of cells with indicated number of centrioles per cell, quantified by scoring centrin-positive foci. $n = 3$ independent experiments with 80 cells measured in each, *$p < 0.034$ vs. RFP + Tet. **b** The FAM46C shRNA centriolar overduplication phenotype is dependent on Plk4 expression, as illustrated by representative immunofluorescence images of U2OS RFP shRNA or FAM46C shRNA cells transfected with Luciferase siRNA or Plk4 siRNA-A X48h (top panels), and labeled with antibodies to centrin (red) and pericentrin (green), and with Hoechst (blue). Inserts to the right show magnified centrosomes (boxed in white) for each condition. Bottom panel: Bar graph showing proportion of cells with indicated number of centrioles per cell, quantified by scoring centrin-positive foci. $n = 2$ independent experiments with >50 cells measured in each. Data are means ± SEM.

FAM46C/TENT5C was recently confirmed as an active non-canonical poly(A) polymerase that enhances mRNA stability and thereby alters gene expression[42]. FAM46C's nucleotidyl transferase activity is dependent on residues D90 and D92, and mutation of these residues to alanine results in abrogation of catalytic activity. Importantly, catalytically inactive mutant FAM46C D90/92A retained the ability to suppress Plk4 activity, similar to wild-type FAM46C (Supplementary Fig. 7b). Furthermore, the ability of FAM46C to suppress Plk4-induced centriole overduplication did not require nucleotidyl transferase activity (Supplementary Fig. 7c). These observations support the hypothesis that FAM46C physically interacts with Plk4 to inhibit its kinase activity, rather than regulating Plk4 and/or centriole duplication through transcriptional regulation.

While the FAM46C sequence contains two consensus Plk4 phosphorylation sites (S74, T354)[50], it did not appear that FAM46C was a substrate for full-length Plk4 (Fig. 4a). However, co-immunoprecipitation experiments using deletion constructs showed that FAM46C interacted with the isolated Plk4 kinase domain, and to a lesser extent with polobox domains PB-1/PB-2, and not with PB3 (Fig. 4d). Of note, FAM46C co-immunoprecipitated with the kinase-dead Plk4 K41M (Supplementary Fig. 8a), implying that the physical interaction was not dependent on Plk4 kinase activity. As expected from its inability to autophosphorylate and induce its own degradation, Plk4 K41M was more abundant intracellularly than wild-type Plk4 (Fig. 4c).

However, there was a limited stabilizing influence of FAM46C on Plk4 K41M (Fig. 4c), again implying that the functional effect of FAM46C depended on Plk4 kinase activity. Trans-autophosphorylation of exogenous kinase-dead Plk4 by endogenous wild-type Plk4 with which it is dimerized has been described[35], and the minor stabilization of Plk4 K41M observed in the presence of RFP-FAM46C may have reflected such homodimerization with endogenous Plk4, the latter being susceptible to regulation by FAM46C. The significant stabilizing effect of FAM46C on wild-type Plk4 protein level could also have potentially been due to an increase in Plk4 mRNA level mediated through FAM46C's poly(A) RNA polymerase activity, but if this were the dominant mechanism, one might expect a corresponding increase in K41M protein stability, which was not observed. The phosphodegron that mediates Plk4 autophosphorylation-dependent degradation was not required for interaction with FAM46C (Fig. 4d), and FAM46C interacted with non-degradable Ser293A/Thr297A mutant Plk4 (Supplementary Fig. 8b). These results argue against the possibility that FAM46C occluded the phosphodegron and suggest instead that direct interaction with the Plk4 kinase domain/ PB-1/PB-2 may be of greatest relevance to the inhibition of Plk4 kinase activity by FAM46C. N terminal/ DUF domain fragments of FAM46C interacted with Plk4, whereas a C terminal fragment did not, and while full-length FAM46C suppressed Plk4 autophosphorylation, the non-interactive C terminal fragment had no apparent effect

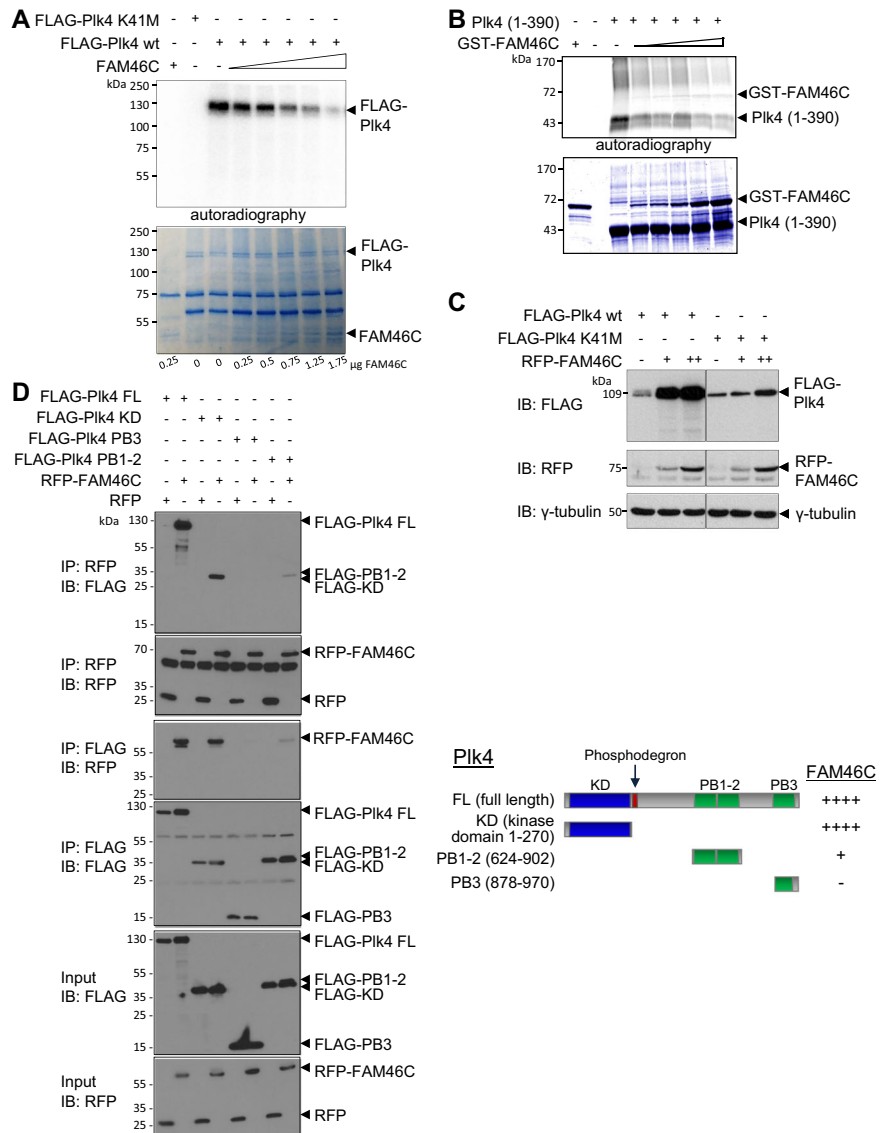

**Fig. 4 FAM46C regulates Plk4 kinase activity. a** In vitro kinase assay showing dose-dependent reduction in wild-type (wt) Plk4 autophosphorylation by FAM46C. Increasing amounts of FAM46C are indicated by wedge, specified for each lane under the colloidal blue-stained gel of input proteins. Autoradiographs show incorporation of $\gamma$-[33]P during incubation, reflecting active phosphorylation. The kinase-dead construct FLAG-Plk4 K41M lacks autophosphorylation. **b** In vitro kinase assay using bacterially expressed purified proteins, showing dose-dependent reduction in Plk4 kinase domain (1–390) autophosphorylation by GST-FAM46C. Increasing amounts of GST-FAM46C are indicated by wedge. **c** Increase in Plk4 protein level in response to forced expression of FAM46C, shown in representative immunoblot of HEK293T cells transfected with FLAG-Plk4 wt or FLAG-Plk4 K41M and increasing amounts of RFP-FAM46C, as indicated, using anti-FLAG and anti-RFP antibodies, with $\gamma$-tubulin as a loading control. The effect on kinase-dead Plk4 protein level was much less pronounced. **d** Immunoblots of RFP-FAM46C and the indicated wild-type Plk4 fragment, after coexpression in HEK293T cells in a reciprocal coimmunoprecipitation assay. Right panel, summary of domain-dependent interactions between Plk4 fragments coexpressed with RFP-FAM46C as in left panel, showing interaction of RFP-FAM46C with the Plk4 kinase domain and PB1-2.

(Supplementary Fig. 8c, d). Interestingly, comparing hFAM46C to paralogs FAM46A, B and D, there is greater divergence in the N terminal versus the C terminal sequence, an observation that, taken together with the failure of the C terminal fragment to bind to or inhibit the kinase activity of Plk4, could underlie the specific interaction of FAM46C with Plk4 (Fig. 1b; Supplementary Fig. 7a).

**FAM46C suppresses the growth of MDA MB-435 cancer xenografts.** Deletion and/or point mutation of FAM46C, found in 25–30% of patients with multiple myeloma, portends an adverse prognosis[51–53]. Forced FAM46C expression suppresses

myeloma cell growth, with altered expression of genes involved in survival signaling pathways[42,54], and faster proliferation of B-lymphocytes harvested from FAM46C knock-out mice. To test the effect of FAM46C on tumor growth in vivo, we employed an aggressive human cancer cell line xenograft model in nude mice. Using two different shFAM46C constructs in MDA MB-435 cells, we confirmed that the resultant tumors retained reduced FAM46C expression in vivo up to 3 weeks after implantation (Fig. 5a). Tumors formed by shFAM46C cells progressed rapidly compared to shLuciferase controls (Fig. 5b–d). Tumors formed by Plk4-depleted MDA MB-435 cells (Fig. 5a) displayed modestly reduced growth (Fig. 5b–d), similar to previous findings in an MDA MB-231 model[19]. The rapidity of tumor growth in the

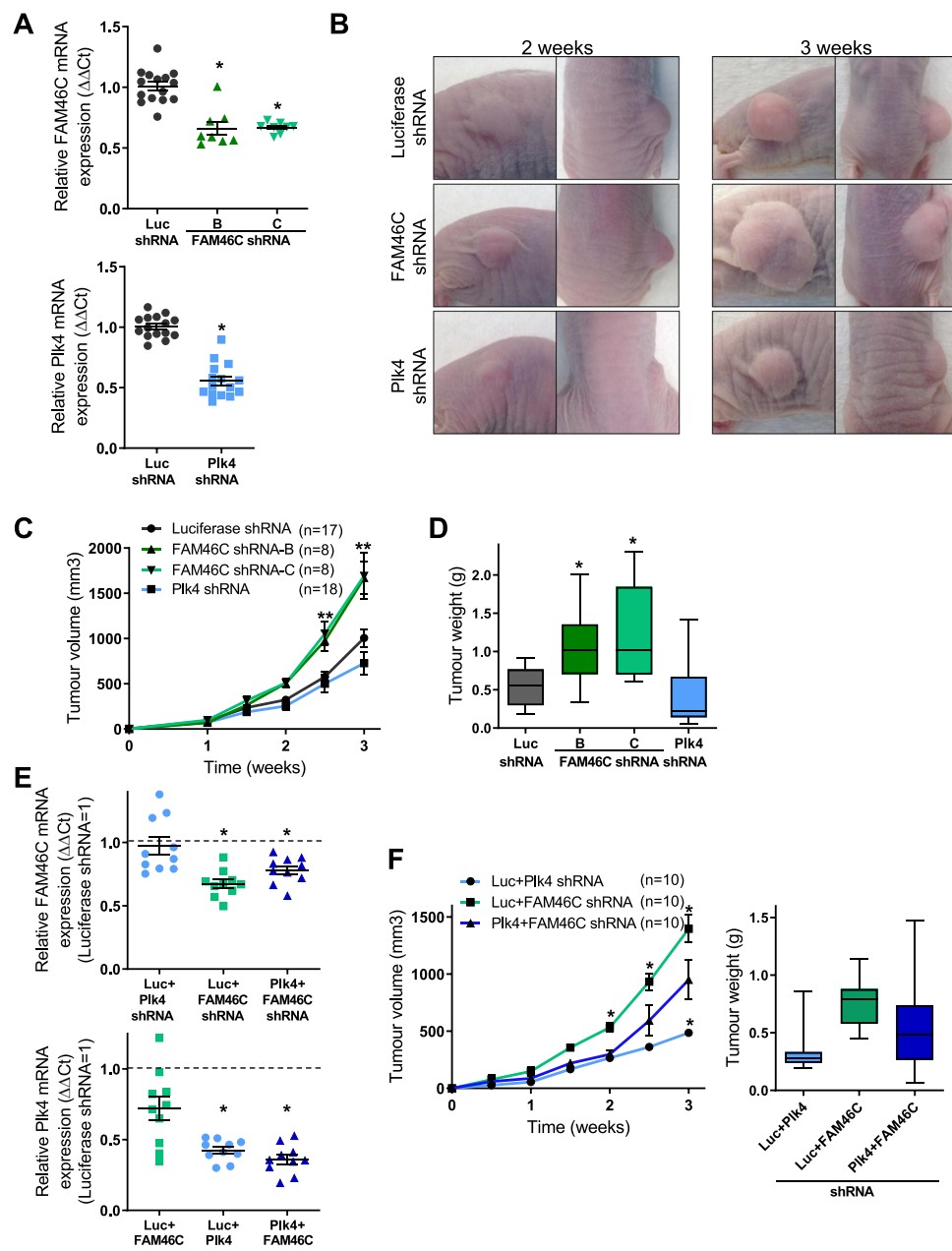

**Fig. 5 FAM46C suppresses MDA MB-435 xenograft tumor growth. a–d** Tumors were generated by injecting MDA MB-435 cells transduced with shRNAs as indicated, and suspended in Matrigel, subcutaneously into the right flank of nude mice, and followed over the ensuing 3 weeks. **a** Quantification of FAM46C and Plk4 mRNA levels in tumors at 3 weeks, showing persistence of FAM46C or Plk4 depletion, as appropriate, relative to control, *$p < 0.001$ vs. Luciferase (Luc) shRNA. Quantification is relative to the housekeeping gene RPII. **b** Representative images of flank tumors in nude mice at the indicated times after injection. **c** Tumor volume of xenografts at indicated times after injection, showing larger tumor size of FAM46C shRNA compared to Luciferase shRNA xenografts at 2.5 and 3 weeks, **$p < 0.012$ vs. Luc shRNA, and showing smaller size of Plk4 shRNA compared to Luciferase shRNA xenografts at 3 weeks. **d** Corresponding tumor weights at time of sacrifice, which was 3 weeks after injection, showing greater weight of FAM46C shRNA xenografts, *$p < 0.008$ vs. Luc shRNA. **e, f** Functional interaction between Plk4 and FAM46C was examined using compound mutant MDA MB-435 cells in a nude mouse xenograft model. **e** Quantification of FAM46C and Plk4 mRNA levels in tumors at 3 weeks after injection, showing successful co-depletion. Dashed line indicates shLuc control, to which other conditions are normalized. Top panel, *$p < 0.019$ vs. Luciferase+Plk4 shRNA. Bottom panel, *$p < 0.001$ vs. Luciferase+FAM46C shRNA. **f** Tumor volume of xenografts at indicated times, showing a partial rescue of the smaller tumor size that occurs with Plk4 depletion alone in Plk4 + FAM46C shRNA xenografts, *$p < 0.033$ vs. Plk4 + FAM46C shRNA. Right panel, corresponding tumor weights showing an intermediate tumor weight in the Plk4 + FAM46C shRNA xenografts vs. Luciferase+Plk4 shRNA and Luciferase+FAM46C shRNA tumors, $n = 10$ mice per condition. Data are means ± SEM.

MDA MB-435 model necessitated early sacrifice, precluding assessment of tissue invasion and distant metastases. Compound mutant cells with knockdown of both Plk4 and FAM46C (Fig. 5e) showed an intermediate phenotype, consistent with a functional interaction in vivo whereby FAM46C restrained the oncogenic effect of Plk4 (Fig. 5f). Of note, there was a trend to reduced Plk4 mRNA in cells depleted of FAM46C, which could potentially reflect poly(A) RNA polymerase activity of the latter. Despite this, depletion of FAM46C enhanced tumor progression (Fig. 5f), suggesting that the dominant effect was to liberate Plk4 kinase

activity. Depletion of Plk4 did not appear to affect FAM46C expression, at least in MDA MB-435 derived xenografts.

**FAM46C is depleted in human colorectal cancer.** Observational studies of clinical cancers and xenografts indicate that high Plk4 expression is associated with aggressive behavior and treatment resistance[8,10,55–57]. For instance, recurrence-free survival was significantly poorer in Tamoxifen-treated breast cancer patients with high vs. low Plk4 expression (42% vs. 66%, GSE6532)[58]. In a cohort of patients with synchronous or metachronous liver metastases from colorectal adenocarcinoma (Supplementary Table 1), we examined the expression of 48 genes in the hPlk4 interactome as defined by mass spectrometry[19] plus 10 additional related genes including FAM46C (Supplementary Table 2), in banked paired samples of primary tumor (T) and adjacent normal mucosa (NM), microdissected to isolate cancer cells and intestinal mucosal cells, respectively. As noted in previous colorectal cancer patient cohorts[8,59], Plk4 was increased in T vs. matched NM (Fig. 6a, b). By contrast, FAM46C expression was consistently reduced in the tumor samples (Fig. 6a, b). This depletion in tumor tissue was not observed for the other centriolar proteins that interact with Plk4 (Supplementary Fig. 9). The ratio of FAM46C to Plk4 expression was markedly reduced in T vs. NM in this patient cohort (Fig. 6c). Analysis of expression array data generated by Kim et al. in an independent cohort of colorectal cancer patients with liver metastases[60] similarly reveals depletion of FAM46C levels relative to Plk4 in colorectal cancer specimens (Fig. 6d). Data from another independent cohort of colorectal cancer patients of all TNM stages[61] show a significant reduction in FAM46C/Plk4 ratio in primary tumor (T) vs. NM (Fig. 6e), and further depletion of FAM46C with clinical stage progression, as apparent in our own initial patient cohort (Fig. 6f). While Plk4 is best recognized for its role in centriole duplication and accurate chromosomal segregation[2,3,62], we and others have demonstrated its promotion of EMT and cell invasion[5,10,19,63]. Here we show that FAM46C suppresses cancer cell invasion in a spheroid model, while FAM46C depletion promotes invasion (Fig. 6g, h). Importantly, when spheroids were treated with the highly selective Plk4 kinase inhibitor centrinone B, the stimulating effect of FAM46C depletion on invasion was lost (Fig. 6h), demonstrating the functional dependence of the latter on Plk4 activity. The causal link between FAM46C and Plk4 demonstrated in these invasion experiments implies that FAM46C can additionally restrain non-centriolar oncogenic functions of Plk4.

## Discussion
The best described and understood role of Plk4 is as a driver of centriole duplication, but other distinct functions have recently been uncovered, including regulation of the actin cytoskeleton with promotion of cell motility[19,64]. Mounting evidence has implicated Plk4 in cancer progression, which can be mediated through Plk4-induced centriolar amplification, amongst other possible mechanisms[5,9,13,16,19,63]. In this study, we show that FAM46C/TENT5C interacts physically with Plk4 in co-immunoprecipitation experiments, and suppresses Plk4 activity. FAM46C restrains centriole duplication and cancer cell invasion in opposition to Plk4, and acts as a tumor suppressor in a human cancer xenograft model. Furthermore, we show that FAM46C is significantly depleted in human colorectal cancer, which correlates with cancer progression. These results reveal FAM46C as a modulator of centriolar duplication via its inhibition of Plk4, and implicate it as a tumor suppressor in common human epithelial malignancies.

Our interest in FAM46C as a potential Plk4 interactor was prompted by the results of yeast 2 hybrid screens, and the specific localization of FAM46C to centrioles we observed here was not anticipated. Taken together with the evidence of physical interaction in vitro, and functional interaction in regulation of centriole duplication and of xenograft tumor growth, the similarity in centriolar localization of FAM46C and Plk4 suggests the possibility of a functional interaction at the centriole; however, further study using super-resolution microscopy will be necessary to elucidate the details of this relationship. The requirements for centriolar localization of FAM46C remain to be determined, but seem unlikely to involve the scaffolding proteins CEP192 or CEP152.

The evidence we have uncovered in the present experiments suggests that binding of FAM46C to Plk4 may occur predominantly within the kinase domain of the latter, but is not dependent on kinase activity, and does not result in phosphorylation of FAM46C. Further investigation of the mechanism of inhibition of Plk4 kinase activity will determine its dependence on direct physical interaction. Analysis of FAM46C/TENT5C sequence and domain structure predicted its function as a non-canonical poly(A) RNA polymerase, and mutation in the polymerase domain renders it catalytically inactive[45]. We found that mutant FAM46C that was catalytically inactive for nucleotidyl transferase activity retained the ability to suppress Plk4 activity, as measured in both kinase and centriole duplication assays. Thus it appears unlikely that the effect of FAM46C on Plk4-dependent centriole duplication is related to its RNA polymerase function, at least in the cancer cell types we have studied here, which include osteosarcoma, melanoma and colorectal adenocarcinoma. In multiple myeloma cell lines, depletion of FAM46C increases proliferation while restoration of functional wild-type FAM46C causes cell death, consistent with the transcriptional regulation of cell survival and death genes observed in those cell lines[54]. In the solid tumor cells studied here, viability was not affected, and alterations in Plk4-dependent centriole duplication antedated changes in proliferation. Moreover, Plk4 mRNA levels were reduced by FAM46C depletion in vivo, the latter nevertheless resulting in enhanced xenograft growth. Taken together, these data are more consistent with unrestrained Plk4 kinase activity as the principal mechanism of tumor promotion, rather than regulation of RNA stability. Inhibition of Plk4 kinase activity and restraint of Plk4-induced centriole overduplication by mutant FAM46C that has no nucleotidyl transferase activity further support the conclusion that the restraining effect on centriole duplication is independent of FAM46C's RNA polymerase function. Stabilization of Plk4 protein levels was dependent on reduced Plk4 kinase activity, rather than increased Plk4 mRNA expression, further implicating suppression of Plk4 kinase activity through direct interaction with the kinase and/or PB-1/PB-2 domains.

Regulation of Plk4 function occurs at both the protein and kinase activity levels, with a complex interplay between the two inherent in the enzyme's autophosphorylation-dependent degradation, and the requirement to achieve a threshold level of Plk4 kinase activity to trigger its self-destruction[32]. Stabilization of wild-type Plk4 protein levels by FAM46C is consistent with its function as an endogenous inhibitor of Plk4 kinase activity. The synthetic Plk4 inhibitor centrinone/centrinone B causes loading of centrioles with inactive wild-type Plk4; withdrawal of centrinone then triggers multiple rounds of unfettered centriole duplication[18]. We found that knockdown of endogenous FAM46C increases centriole duplication in a Plk4-dependent manner, creating centriolar rosettes characteristic of forced Plk4 expression or activation. There is no apparent structural similarity between FAM46C/TENT5C and any of the pharmacologic

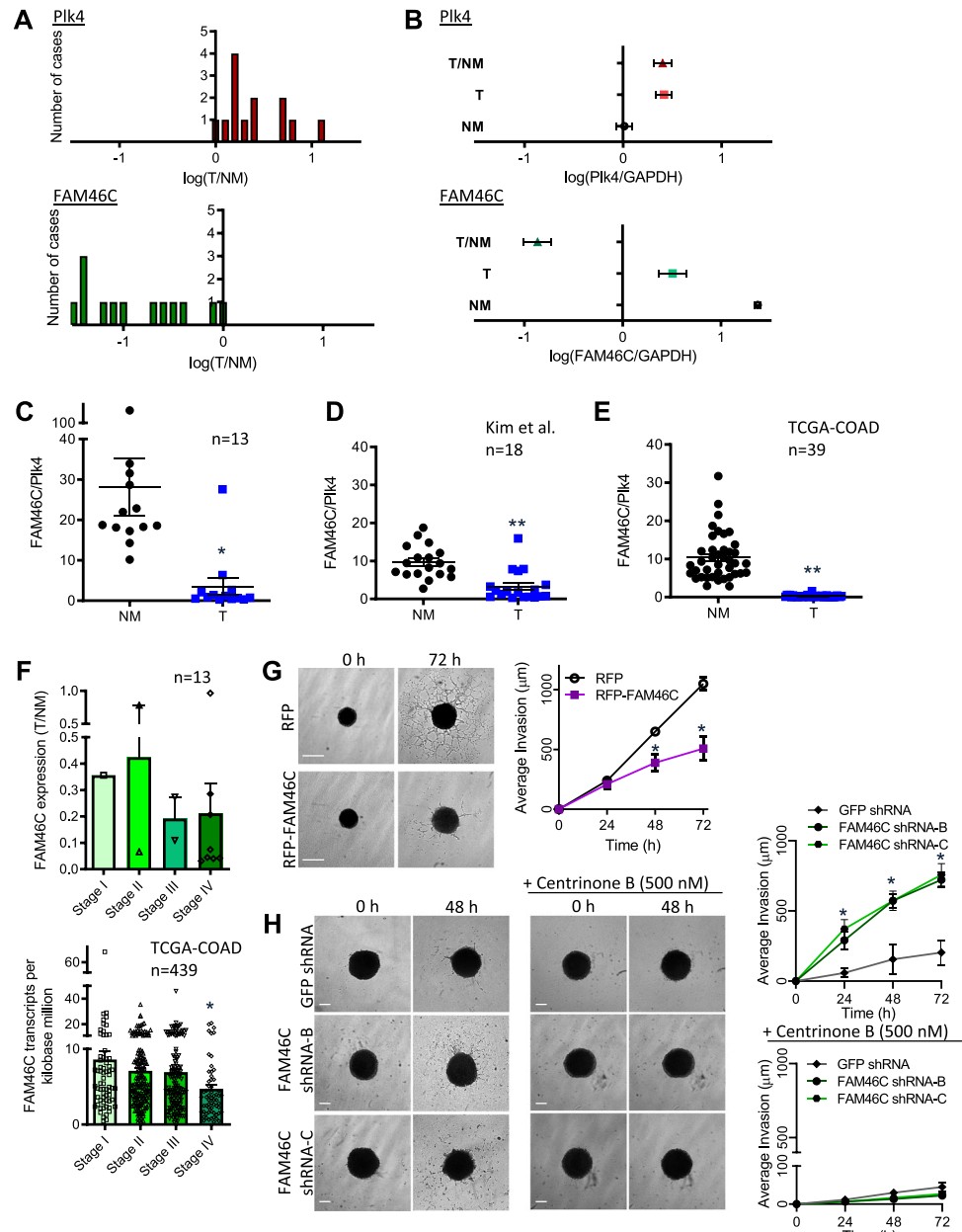

**Fig. 6 FAM46C is depleted in human colorectal cancer. a** Distribution of Plk4 (top panel) and FAM46C (bottom panel) expression levels in colorectal primary tumor (T) compared with paired normal mucosa (NM) in microdissected specimens from 13 cases of advanced colorectal cancer. T/NM ratio is displayed on a log scale. Plk4 and FAM46C expression were measured relative to the control GAPDH. Each patient case is represented by a single vertical unit. All patients had synchronous or metachronous liver metastases; this cohort is described in further detail in Table S1. **b** Summary of expression levels of Plk4 (top) and FAM46C (bottom) in 13 cases of colorectal cancer, as in **a**, showing increased Plk4 and decreased FAM46C in primary tumor (T) vs. paired normal colonic mucosa (NM). Data are log-transformed means ± SEM. **c–e** Ratio of FAM46C/Plk4 expression in colorectal primary tumor (T) vs. paired normal mucosa (NM), assayed using qPCR (**c** *$p < 0.003$, vs. NM, $n = 13$ patient cohort as in **a** above), or derived from RNA-seq data of Kim et al., 2014[60] (**d** **$p < 0.0001$, vs. NM, $n = 18$), or The Cancer Genome Atlas (TCGA-COAD), 2012[61] (**e** **$p < 0.0001$, vs. NM, $n = 39$). **f** Top panel, FAM46C expression in primary tumor (T)/paired normal mucosa (NM) in 13 patients with colorectal cancer, as in **a** above, sorted by stage at time of primary presentation: Stage I, $n = 1$; Stage II, $n = 2$; Stage III, $n = 2$; Stage IV, $n = 8$. Bottom panel, FAM46C expression in colorectal primary tumor determined by RNA-seq, in a cohort of 439 patients, from TCGA-COAD, 2012[61], sorted by stage: Stage I, $n = 74$; Stage II, $n = 175$; Stage III, $n = 128$; Stage IV, $n = 62$. *$p < 0.005$ vs. Stage I. **g, h** Representative brightfield images of HeLa cells plated on ultra-low attachment plates, which form spheres over 4 days in culture after addition of Matrigel. **g** Transfection with RFP-FAM46C suppressed the 3D invasion of cells into surrounding matrix, at the indicated times. Quantification of invasion is shown in right panel, $n = 8$ independent experiments, *$p < 0.01$ vs. RFP alone. **h** FAM46C knockdown using shRNA enhanced the invasion of spheroids into surrounding matrix. Synchronous treatment with the selective Plk4 inhibitor centrinone B prevented the enhanced invasion seen with FAM46C knockdown, indicating that the enhancement is dependent on Plk4 activity. Quantification of invasion is shown in right panels, $n = 7$ independent experiments for each panel, *$p < 0.001$ vs. GFP shRNA without centrinone B (top panel) and p = NS with centrinone B (bottom panel). Data are means ± SEM. Bars: 300 μm.

Plk4 inhibitors previously described, which in addition to centrinone/centrinone B also include CFI-400945[9,21] and YLT-11[65]; each of these agents was designed to block the ATP-binding pocket of Plk4.

Regulation of Plk4 function can also be mediated through its most intimate functional interactors, such as STIL. Binding of CDK1-CyclinB to STIL during mitosis prevents its binding to Plk4, delaying phosphorylation of STIL by Plk4, and the resulting recruitment of SAS-6 and triggering of centriolar biogenesis, until G1[66]. Of note, phosphorylated STIL enhances the centriolar anchoring of Plk4; the normal restriction of procentriole formation to the single spot of Plk4 concentration on the mother centriole is disrupted by overexpression of either STIL or Plk4. While we have shown here that FAM46C physically interacts with Plk4 within its kinase/ PB-1/PB-2 domains, suppressing its autophosphorylation, the impact on Plk4-STIL interaction remains to be determined. However, depletion of endogenous FAM46C does culminate in enhanced recruitment of endogenous SAS-6 to procentrioles, implying that activity of the Plk4-STIL axis is indeed upregulated.

Loss or mutation of FAM46C is a common secondary genetic event in multiple myeloma, and predicts inferior survival[51–54]. The reason(s) for this association remained obscure while the function of FAM46C was unknown. Members of the FAM46 family were long speculated to function as RNA polymerases, but definitive evidence of nucleotide transferase activity was uncovered only recently[45], in recognition of which the FAM46C gene was renamed terminal nucleotidyl transferase 5C (TENT5C). It regulates gene expression through polyadenylation-induced enhancement of mRNA stability. As for the specific mRNAs targeted by FAM46C-induced polyadenylation, in multiple myeloma lines these were shown to be highly enriched for proteins that are targeted to the ER[42]. Mroczek et al. propose that loss of FAM46C is associated with ER stress in multiple myeloma, making this tumor type particularly vulnerable to proteasome inhibition. The role of FAM46C/TENT5C poly(A) polymerase activity in other types of cancer requires further study. Previous reports of FAM46C depletion in hepatocellular cancer have suggested an association with tumor progression and adverse patient prognosis[67], foreshadowing our present analysis of FAM46C in colorectal cancer. These prognostic correlations could be related to altered expression of multiple genes involved in cell survival/death, or, as we suggest here, to unopposed oncogenic kinase activity of Plk4. It should be acknowledged that these are not mutually exclusive mechanisms. Intriguingly, the anti-proliferative and anti-invasive effects of norcantharidin on hepatoma cells depended on its specific upregulation of FAM46C expression[67,68]. Lack of FAM46C/TENT5C function, now revealed as a frequent and prognostic feature in human colorectal cancer as well as in multiple myeloma and hepatocellular carcinoma, can promote tumor progression by allowing unrestrained Plk4 activity. Restoration of FAM46C function warrants investigation as a therapeutic strategy in treatment-resistant cancers driven by Plk4.

## Methods

**Phylogenetic tree**. Based on multiple sequence alignments generated by Clustal Omega (https://www.ebi.ac.uk/Tools/msa/clustalo/) as in Supplementary Fig. 1, an unrooted phylogenetic tree was generated for select FAM46C orthologs and the human FAM46C paralogs, FAM46 A, B and D.

**Cell culture, transfection**. Cells were grown at 37 °C in Dulbecco's modified Eagle Medium (DMEM: HEK293T), Roswell Park Memorial Institute medium-1640 (RPMI: MDA-MB-435), DMEM-F12 (RPE-1) or McCoy's 5A medium (U2OS) supplemented with 10% fetal bovine serum (FBS). Transient transfection was performed using PEI transfection reagent (Sigma), Lipofectamine 2000 or Lipofectamine RNAiMAX (Invitrogen) according to manufacturer's instructions and as

described[19]. A pool of four FAM46C siRNAs was utilized (M-020700 SMARTpool siGENOME siRNA, Dharmacon). U2OS cells were incubated with 8 mM hydroxyurea for 16 h to effect cell cycle arrest. U2OS Flp-In T-Rex YFP-Plk4 cells were treated with Tetracycline at a concentration of 1 µg/mL X16-24h to stimulate YFP-Plk4 expression. Following 24 h of tetracycline induction, cells were transfected with RFP, RFP-FAM46C, RFP-FAM46C D90/D92A x42h and fixed and stained with centriole markers for immunofluorescence.

**Cell line derivation**. HEK293T, RPE-1 and MDA MB-435 cell lines were a kind gift from the Tony Pawson laboratory (Lunenfeld Tanenbaum Research Institute, Toronto), and U2OS cells were a kind gift from the Laurence Pelletier laboratory (Lunenfeld Tanenbaum Research Institute).

**Stable cell lines**. Stable cell lines were generated as Flp-In U2OS T-REx cells, or U2OS and MDA MB-435 cells expressing Plk4 (SHCLNG-NM_014264, Sigma), Luciferase (SHC007, Sigma), RFP (Tony Pawson laboratory, Lunenfeld Tanenbaum Research Institute) or FAM46C (SHCLND-NG_017709, Sigma) short hairpin RNAs (shRNAs) through lentiviral infection as described[19]. Stable cells transduced with shRNAs were studied within 3 weeks of infection. For transfection in the rescue protocol, 100,000 U2OS FAM46C shRNA (two shRNA constructs utilized) or RFP shRNA cells were seeded onto 6-well plates and transfected after 24 h with 1 µg of RFP-FAM46C or RFP X42h. RFP-FAM46C and RFP expression were confirmed with immunofluorescence. Cells were tested for mycoplasma contamination.

**Plasmid constructs**. Vectors were constructed using Invitrogen's Gateway system as described[19]. The Flag-CEP192 (1–2436), Flag-CEP135 and BirA*-Flag-CEP152 vectors were kindly provided by Laurence Pelletier's laboratory (Lunenfeld Tanenbaum Research Institute, Toronto, Canada). The Plk4 Non-Degradable (ND) mutant, with two mutations within the *DSGIIT* degron (Ser293A and Thr297A), and FAM46C catalytically inactive mutant (D90A and D92A), were synthesized from the PLK4 and FAM46C entry vectors (Gateway) using PCR-based site-directed mutagenesis (QuikChange II Site-Directed Mutagenesis Kit, Agilent Technologies). The Plk4 ND and FAM46C D90/92A mutants were cloned into the pDEST N-terminal 3xFlag vector. Plk4 deletion mutants were cloned as described[19]. FAM46C deletion mutants (N-term 1–199, C-term 193–391 and DUF 17–336) were generated by PCR using Phusion High Fidelity DNA Polymerase (M0530, New England Biolabs) according to manufacturer's instructions.

FAM46A, B, C and D entry vectors (Gateway) were obtained from the Lunenfeld Tanenbaum Research Institute OpenFreezer reagent repository and cloned into the pDEST mCherry destination vector (Gateway). All constructs were validated by sequencing.

**Co-immunoprecipitation, immunoblotting**. These were performed as described[19]. In brief, HEK293T cells transfected using PEI transfection reagent (Sigma) X24h were lysed using TNTE lysis buffer (2 mM Tris-HCl,pH7.5, 120 mM NaCl, 1% TritonX-100, 1 mM EDTA), with protease inhibitor cocktail, 5mMNaF and 2mMNaOva. Beads were pre-washed and blocked with 5% BSA. Extracts were centrifuged (14,000 rpm) X10min, and supernatants immunoprecipitated with anti-FLAG M2 affinity gel (Sigma; A2220) X1.5 h, or incubated with rabbit polyclonal mCherry, RFP or GFP antibodies (Abcam) X1h followed by immunoprecipitation with ProteinG Sepharose beads (GE Healthcare) X30min. Beads were washed 6x with lysis buffer supplemented with 500 mM NaCl, boiled in Laemmli sample buffer, and analyzed using SDS polyacrylamide gel electrophoresis (SDS-PAGE) followed by immunoblotting.

**Anti-FAM46C antibody production**. Full length FAM46C was cloned using the pET system into a bacterial expression vector containing a C-term 6×His tag. Recombinant protein was purified from *Escherichia coli* using Ni-NTA beads (Novagen) and used for immunization; rabbit immune sera were affinity-purified using standard procedures (Pacific Immunology Corp.). Prior incubation of anti-FAM46C with antigen eliminated centriolar staining, and staining was markedly reduced by depletion of FAM46C using siRNA (Fig. 1d).

**Immunofluorescence**. Cells were fixed, permeabilized and blocked using Methanol (−20 °C, 10 min), 0.5% Nonidet-P40 (Bioshop, 20 min), and 0.2% Fish Gelatin/PBS X1h. Antibody incubations were in the blocking solution overnight (4 °C), and slides were mounted in Immuno-mount medium (Thermo). Immunofluorescence images were collected using the Olympia Deconvolution fluorescence microscope or Deltavision Elite DV imaging system equipped with a sCMOS 2048×2048 pixels2 camera (GE Healthcare). Z stacks (0.2 µm apart) were used, and images were deconvolved and maximum intensity projected using softWoRx software (Applied Precision). Images were collected using 60X and 100×1.4 NA oil objectives (Olympus). High- and super-resolution imaging were performed following standard procedures[69] at the Network Biology Collaborative Centre (NBCC), a facility supported by Canada Foundation for Innovation, the Ontarian Government, and Genome Canada and Ontario Genomics Institute (OGI-139).

**Antibodies**. Antibodies used for immunofluorescence in this study were FAM46C (generated for this study, as above, 1:200), centrin (clone 20H5, Millipore, 1:1000), pericentrin (Sigma, 1:1000), Plk4 (NB100–894, Novus Biologicals Canada and MABC544, Millipore, 1:300), SAS-6 (sc-82360, Santa Cruz, 1:400), CPAP[70] CP110 (A301-343A-1, Bethyl Laboratories, 1:1000), Cep135[70], CEP120[70] (kind gift of Dr. M. Mahjoub, Washington University, St. Louis, 1:3000), ODF2 (H00004957-M01, Cedarlane/Abnova, 1:100). Secondary antibodies were conjugated to Alexa Fluor 488, 546, 594, 633, or 647 (Life Technologies, 1:800). DNA was detected using Hoechst. YFP and mCherry/RFP were visualized directly. The antibodies used for immunoblotting were: anti-β-tubulin (Sigma-Aldrich, 1:1000), anti-γ-tubulin (Sigma-Aldrich, 1:1000), anti-FLAG M2 (F1804, Sigma-Aldrich, anti-RFP (Abcam, 1:1000), anti-mCherry (Abcam, 1:500), anti-FAM46C (generated for this study, as above, 1:1000), anti-GFP (ab290, Abcam, 1:1000), anti-Cyclin B1 (4135, Cell Signaling, 1:1000), anti-Cyclin D1 (2926, Cell Signaling, 1:1000), anti-Phosphohistone H3 (9701, Cell Signaling, 1:1000).

**RNA extraction, real-time RT-PCR**. These were performed as described[19]. In brief, RNA was isolated using the RNeasy mini kit (QIAGEN), treated with RNase-free DNase (Invitrogen) and reverse transcribed with SuperScriptII reverse transcriptase (18064-014, Invitrogen) using Random Primers (48190-011, Invitrogen). Real-time RT-PCR was performed using SYBR Green PCR Master Mix (Applied Biosystems) on an ABI 7900HT apparatus. Quantifications were normalized to control endogenous GAPDH, with data generated by PCR software (SDS2.2.2, Applied Biosystems) analyzed using the $2^{-\Delta\Delta Ct}$ method. Primers for FAM46C were as follows, F: ggccacgttttggtcaaagac and R: gggaacacagaaccacatctc.

**Cell cycle analyses**. HeLa cells were grown to 70% confluence and treated with 2 μg/mL aphidicolin x24h, washed then treated again with 2 μg/mL aphidicolin for another 24 h. Cells were then washed with PBS, and DMEM + FCS added for release. At the indicated times after release cells were lysed and collected for Immunoblotting or cells grown on slides were fixed and stained with centriole markers for Immunofluorescence. In other cell cycle analyses, RPE-1 cells were grown to 70% confluence and treated with 2 mM thymidine (Sigma) X20h, washed with PBS and DMEM-F12 + FCS added for release. After 8 h cells were again treated with 2 mM thymidine X15h and again released. At the indicated times after release cell pellets were collected, fixed in cold 70% ethanol, washed with PBS and resuspended in PBS supplemented with 0.5 mg/ml RNase A (30 min, 37 °C). Propidium iodide (Invitrogen) was added to a final concentration of 10 μg/mL (15 min, room temperature) followed by analysis with a flow cytometer (BD Cytoff). Data were processed using Flow Jo software. For cell cycle immunoblot analysis, cells were harvested with Laemmli buffer and boiled (5 min, 100 °C). Lysates were run on 12% acrylamide SDS gels and transferred onto nitrocellulose (PALL Life Sciences BioTrace) using a BD Transfer Blot SD. Blots were blocked in Licor Blocking Buffer. Primary antibodies in 1:1 Licor Blocking buffer: PBS with 0.1% Tween20 (PBS-T) were incubated overnight at 4 °C. Blots were washed in PBS-T. Blots were then incubated in LICOR secondary fluorescent antibodies (1:30,000) in 1:1 Licor Blocking buffer: PBS-T X1h. Blots were washed in PBS-T and visualized using the LICOR scanner.

**Electron microscopy**. For electron microscopy, HEK293T samples were fixed in 2.5% glutaraldehyde solution buffered with PBS for 2 h. After they were washed three times in phosphate buffer, cells were postfixed with 1% glutaraldehyde for 1 h, then washed in buffer and distilled water. Cells were then dehydrated in a graded series of ethanols and flat-embedded in a mixture of Epon and Araldite. After polymerization for 48 h at 60 °C, coverslips were immersed in liquid nitrogen and sections were obtained with an ultramicrotome. Sections were stained with uranyl acetate and Reynold's lead citrate. Images were obtained with a Philips CM10 electron microscope.

**Xenograft studies in mice**. All protocols were approved by the Toronto Centre for Phenogenomics (TCP) Animal Care Committee. $0.5 \times 10^6$ MDA MB-435 Luciferase, Plk4 or FAM46C shRNA cells were injected subcutaneously in the right flank of female 5wk-old NCr Nude mice (Taconic Biosciences). Tumor growth was monitored by palpation, size was measured with calipers, and volume calculated assuming an ellipsoid shape. Mice were sacrificed at 3wk post-injection. Tumors were harvested and frozen in liquid nitrogen for RNA extraction.

**Real time RT-PCR of xenograft lysates**. Tissue was placed in Ambion RNAlater-ICE solution overnight at –20 °C, then disrupted and homogenized in RLT buffer (RNeasy mini kit Qiagen) supplemented with β-mercaptoethanol using a rotor-stator homogenizer. Lysates were centrifuged X10min (4 °C; 14,000 rpm) and the supernatant used for RNA purification using the RNeasy mini kit following manufacturer's protocol. Real time RT-PCR was performed using SYBR Green PCR Master Mix (Applied Biosystems) on an ABI 7900HT apparatus, and normalized to RPII.

**qPCR**. qPCR reactions were prepared with TaqMan Universal PCR Master Mix no AmpErase UNG (4324018, Applied Biosystems) and TaqMan Gene Expression

Assays 20X Hs00214530_m1 (FAM46C) and Hs00179514_m1 (PLK4), and run on the Applied Biosystems 7900HT instrument. Conditions for the reactions were according to TaqMan protocol. GAPDH was used as a housekeeping gene. For each gene, quantities were extrapolated from standard curves made with control cDNA prepared from a pool of cells. Expression for each gene was calculated as a ratio relative to GAPDH, and primary expression was compared to normal for each individual patient.

**In vitro kinase assay**. HEK293T cells were transfected with FLAG-Plk4 WT, kinase-dead FLAG-Plk4 K41M, FLAG-FAM46A, FLAG-FAM46C WT, FLAG-FAM46C D90/92 A or FLAG-FAM46C 193–391 using PEI transfection reagent (Sigma). After 24 h, cells were lysed using TNTE lysis buffer (20 mM Tris-HCl, pH 7.5, 120 mM NaCl, 1% Triton X-100, 1 mM EDTA), with protease inhibitor cocktail, 5 mM NaF and 2 mM NaOva phosphatase inhibitors. Extracts were centrifuged at 14,000 rpm x 10 min (4 °C), and the supernatants were immuno-precipitated with anti-Flag M2 affinity beads (Sigma; A2220) for 1 h (4 °C). The beads were washed three times with lysis buffer supplemented with an additional 500 mM NaCl and twice with kinase buffer (25 mM Tris HCl, pH 7.5, 25 mM MgCl2, 15 mM sodium glycerolphosphate, 0.5 mM NaOva, 2 mM EDTA, 25 mM NaF, 1 mM DTT and 1.25 μg BSA). The beads were then incubated with 0.25–2.25 μg of FAM46C recombinant protein (Origene), or with eluted protein FAM46A, FAM46C WT, FAM46C 193–391 or FAM46C D90/92A, in 30 μl kinase buffer containing 100 μM ATP with 10 μCi [γ-33P or γ-32P] ATP. For elution of FLAG-FAM46A, FLAG-FAM46C WT, FLAG-FAM46C 193–391 or FLAG-FAM46C D90/92A, the beads were washed three times with lysis buffer, and protein eluted with 15 μg 3XFLAG-Peptide (APExBIO,A6001) with gentle mixing X30 min. Kinase reactions were performed at 30 °C for 30 min and terminated by adding Laemmli sample buffer. Proteins were separated by SDS–PAGE gel electrophoresis, stained with Colloidal Blue (Invitrogen), and dried using a Bio-Rad gel dryer. Phosphorylation was visualized by autoradiography (Typhoon FLA 9500, GE Healthcare).

For experiments using bacterially expressed and purified recombinant human GST-FAM46C and His-Plk4 (1–390), constructs were expressed in BL21 *E. coli* and purified using glutathione resin (NEB) or HisPur Ni-NTA resin (Thermo Fisher Scientific), according to manufacturers' instructions. In vitro kinase reactions were performed using kinase buffer containing 100 μM ATP with 10 μCi [γ-33P] ATP as above, with protein levels determined by Coomassie blue staining.

**3D spheroid invasion assay**. HeLa cell spheroids, untreated or treated as indicated, were generated by seeding 2500 or 2800 cells into Corning Costar Ultra-Low Attachment 96-well plates (7007, Corning) in DMEM + 10%FCS and incubating at 37 °C. Transient transfection with RFP or RFP-FAM46C was performed for 6 h on day 3 using Lipofectamine 2000 (Invitrogen). Matrigel matrix (354234, Corning) was dispensed into wells on day 4 to initiate invasion ($t = 0$ h). Centrinone B (5690, TOCRIS Bioscience), dissolved in DMSO and used at a final concentration of 500 nM, was dispensed into selected wells after solidification of Matrigel. Spheroids were imaged at 24-hour intervals using the In Cell Analyzer 6000 (GE Healthcare Life Sciences), equipped with a Nikon 4X/0.20 NA, Plan Apo objective and 2048×2048 sCMOS camera. Quantification was performed by outlining invasion on the ImageJ software, then calculating average distance of invasion from the center of the spheroid using the following equation:

$$\text{Average invasion (μm)} = (\text{x axis length}/2 + \text{y axis length}/2)/2$$

The outer perimeter of the invadopods extending into the Matrigel matrix was outlined and two measures each of the distance through the center of the spheroid in the x and y axes were taken and averaged to determine degree of invasion.

**Colorectal cancer specimen tissue processing, microdissection, RNA isolation, and pre-amplification**. All protocols were approved by the Sinai Health Systems and University Health Network Research Ethics Boards. Fresh frozen pairs of primary colorectal cancer tumors and corresponding liver metastases, along with normal colonic tissue, were obtained from the University Health Network and Lunenfeld Tanenbaum Research Institute BioBanks. Sections on RNAse free PET membrane slides (11505190, Leica Microsystems) were stained with hematoxylin and eosin according to standard protocols, washed with RNAse free water then laser capture microdissected using the Zeiss PALM MicroBeam (ZEISS) to isolate strictly tumor tissues from each frozen section. RNA from microdissected tissue was isolated using the PicoPure RNA isolation kit (KIT0204, Arcturus), including DNAse treatment (79254, RNase-Free DNase Set, Qiagen). Normal colonic mucosal tissue was placed in Ambion RNAlater-ICE solution overnight at –20 °C, then disrupted and homogenized by rotor/stator in buffer XB with β-ME added (5 μl β-ME /500 μl buffer). RNA was isolated using PicoPure RNA isolation kit, including DNase treatment. RNA was quantified using Nano-Drop. 15 ng RNA for each sample was reverse-transcribed using SuperScript IV First Strand Synthesis System (18091050, Invitrogen) and random hexamers. cDNA was preamplified with TaqMan PreAmp Master Mix (4391128, Applied Biosystems) and the pooled assay mix (0.2X each assay), and then the preamplified cDNA was diluted as recommended.

**TCGA-COAD**. Transcriptome and clinical datasets from TCGA-COAD[61] and Kim et al.[60] were downloaded using the NIH National Cancer Institute GDC Data Portal Data release 8.0 https://portal.gdc.cancer.gov/, cBioPortal for Cancer Genomics v1.8.3 http://www.cbioportal.org/ [71,72] and CRN Nexus http://syslab4.nchu.edu.tw/ [73].

**Sucrose gradient**. Isolation of centrosomes was performed by sucrose gradient density centrifugation[74]. 70% confluent U2OS cells were co-transfected with FLAG-Plk4 and RFP-FAM46C X24h. Cells were lysed and protein extracts separated through a sucrose gradient by ultracentrifugation at 10,400 G for 30 min to sediment the proteins on a sucrose cushion and at 11,2700 G X1h for final centrosomal purification. The gradient was fractionated from the bottom, yielding fourteen sucrose fractions that were precipitated with ethanol and chloroform, then subjected to SDS/PAGE followed by immunoblotting.

**Statistics and reproducibility**. Statistical significance and $p$ values were assessed by analysis of variance with Bonferroni correction and Student's $t$ tests using Prism software (GraphPad Software, La Jolla, CA). Error bars reflect SEM. Number of replicates and samples sizes are stated in the respective Figure Legend for each figure, with n value corresponding to independent experiments.

**Reporting summary**. Further information on research design is available in the Nature Research Reporting Summary linked to this article.

## Data availability
The datasets generated and/or analyzed as part of the current study are available from the corresponding author upon reasonable request. Source data are made available as "Supplementary Data 1".

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

## Acknowledgements

The authors acknowledge Laurence Pelletier, Jim Dennis, Anne-Claude Gingras, Julie Brill, Gabrielle Boul, and Johnny Tkach for helpful discussions, and Mona Reid and Johnny Tkach for technical assistance. This work was supported by a grant from The Cancer Research Society (CJS) and by the Sydney Cooper Program for the Prevention of Cancer Progression, Mount Sinai Hospital, Toronto, Canada.

## Author contributions

K.K., Y.H., and C.J.S. conceived of the study and K.K. and Y.H. conducted the majority of the experiments. K.K. and C.J.S. wrote the manuscript, to which W.J. contributed. C.M. M.L. and D.N. conducted the spheroid assays, to which R.X. contributed, and D.N. performed the sucrose gradient centrifugation. M.K. performed laser capture micro-dissection, enabled by F.S.W.Z., and A.P. identified tumor tissue and normal mucosa in colorectal cancer specimens. J.T. and K.P. contributed to the mouse xenograft experiments. A.S. performed the cell cycle analysis in RPE-1 cells. H.W. conducted histologic examination of murine xenograft tumors.

## Competing interests

The authors declare no competing interests.
