## [Peer Review File · Communications Biology]

Reviewers' comments:

Reviewer #1 (Remarks to the Author):

In this manuscript, Kazazian and colleagues identified a new regulator, FAM46C, for centriole duplication in human cells. Based on the previous Y2H data, the authors tested the localization of FAM46C as a potential interactor for Plk4, and found that FAM46C is indeed a centriole protein that physically interacts with Plk4. It seems that FAM46C colocalized with Plk4 at centrioles. The specificity of the interaction between Plk4 and FAM46C was also addressed with FAM46 family proteins. Reduction of FAM46C expression resulted in an increase of centriole number, which was suppressed by concomitant knock-down of Plk4. Consistent conclusion was drawn from the experiment in which overduplication of centrioles by Plk4 overexpression was suppressed by FAM46C overexpression. To gain insights into more direct regulation of Plk4 by FAM46C, the authors performed in vitro kinase assays with purified Plk4 and FAM46C proteins. This experiment revealed that FAM46C inhibits the kinase activity of Plk4 through its interaction with the kinase domain. The functional relationship between Plk4 and FAM46C was further addressed by using the xenograft tumor growth. In addition, loss of FAM46C was more frequently observed in colon cancer patient tissues with advanced clinical stages.

Overall, this reviewer could not be enthusiastic for publication of this manuscript in *Communications Biology*, since the experimental evidences to support the function of FAM46C regulating Plk4 activity are rather weak and too preliminary. In particular, the biochemical data are not compelling and not analyzed in a quantitative manner. Although the significance of FAM46C for centriole duplication is clear, its mechanism proposed in this study is not convincing. Thus, although some pieces of the data are potentially interesting, this manuscript may be better suited for a more specialized journal.

(Major points)

1. In Figure 1C, the exact localization of FAM46C at centrioles should be addressed at the super-resolution level. Plk4 is known to localize at the proximal end of procentrioles. Is it also true for FAM46C? The localization should be also observed across the cell cycle.
2. In Figure 1 and 2, Centriole amplification upon depletion of FAM46C is interesting. The authors should test whether it is because of centriole reduplication or multiple daughter centriole formation against one mother centriole, by using mother centriole markers such as Cep164, C-Nap1 or ODF2.
3. In Figure 1B, Would the interaction between Plk4 and FAM46C need the NTase activity of FAM46C? The requirement of the NTase activity in the regulation of centriole duplication should be also tested by rescue experiments.
4. In Figure 2, is there a way to monitor the Plk4 activity in the cell expressing FAM46C, to test whether Plk4 activity is indeed enhanced at centrioles? Also, did the authors check the expression level of Plk4 at centrioles in the absence of FAM46C?
5. In Figure 3, it would be more informative to clarify which region is needed for the ability of FAM46C to suppress centriole amplification by Plk4 overexpression.
6. In Figure 4A-B, these data are not convincing because the specificity of inhibition of Plk4 kinase activity by FAM46C is not clear. For example, the amount of FAM46A added into the reactions seems not to be proper based on the intensity of blue-stained bands.
7. How do the authors exclude the possibility that the protein amount of Plk4 or its regulators, rather than the Plk4 kinase activity, is regulated by FAM46C at the transcription level?

Reviewer #2 (Remarks to the Author):

In their manuscript Kazazian and coworkers describe FAM46C as a novel regulator of PLK4-induced centriole replication. They find that FAM46C localizes to centrioles and interferes with PLK4 kinase activity by binding to its kinase domain. In addition to inhibition of centriole duplication, FAM46C suppressed the growth of a breast cancer cell line in nude mice and invasion of HeLa cells in a sphere culture model system. These findings are novel and of interest to the scientific community. Overall, the manuscript is well written and the data presented are convincing. Nevertheless, I have some comments:

1. Figures 1C and 1D: What is the percentage of cells showing centrosomal immunofluorescence signals after FAM46C antibody staining (Fig. 1C) and transfection with RFP-FAM46C (Fig. 1D)? Is the centrosomal localisation of these signals cell cycle dependent?
2. Centrosomal localization of FAM46C should in addition be confirmed at the biochemical level by Western blotting of centrosomal extracts.
3. The example images given in Figure 2B do not exactly resemble centriole rosettes. To prove that the structures really resemble bona fide rosettes, a co-immunostaining with antibodies that allow for discrimination between mother/parental and daughter centrioles should be performed.
4. Figures 2D and 3A: The authors show here that transient transfection with RFP-FAM46C leads to reduced centriole numbers in U2OS cells as well as U2OS T-REx YFP-Plk4 cells 42 h and 40 h after transfection, respectively. As transient transfection causes G1 cell cycle arrest in many cells, this arrest might contribute to reduced centriole replication. Inducible, stably transfected RFP-FAM46C cells would solve this problem.
5. Figure 6A: Depicting the tumor/normal mucosa ratios of Plk4 and FAM46C on a log scale seems misleading to me. They should be given on a linear scale as well.
6. As FAM46C mutations have been described to occur and influence prognosis in multiple myeloma, experiments performed in that cancer entity might strengthen the data presented.

Reviewer #3 (Remarks to the Author):

The authors mined interactome databases to find that only four Plk4-interacting proteins were shared across species. Among these, the authors found that FAM46C localizes to centrioles where it partially overlaps with Plk4. FAM46C-depletion led to a supernumerary centriole phenotype reminiscent of that caused by Plk4 overexpression, while overexpression of FAM46C led to a decrease in centriole number reminiscent of Plk4 depletion, suggesting that FAM46C acts at centrioles by inhibiting Plk4 function. To validate this suggestion, the authors demonstrate reciprocal co-IP between FAM46C and Plk4, and show that the supernumerary centriole phenotype of FAM46C-depletion is dependent on Plk4, while overexpression of FAM46C can prevent the production of excess centrioles driven by overexpression of Plk4. They also show that FAM46C inhibits Plk4 autophos in vitro and increases Plk4 stability in cells (presumably due to attenuation of the autophos-driven degradation of Plk4). The authors then explore the potential role for FAM46C in tumorigenesis, finding that depletion of FAM46C enhances tumorigenesis of MDA-MB-435 xenografts in mice, that a reduction in FAM46C expression correlates with disease progression in human colorectal tumors, and that FAM46C overexpression reduced invasive activity of cancer cells from spheroids into surrounding matrigel. Together, these

observations support the suggestion that FAM46C is a tumor suppressor that acts by inhibiting Plk4-dependent centriole biogenesis.

Overall, this is a very strong study that will be of great interest to the centriole field, and of general interest to cell biologists. The data are of high quality, and the authors have pursued multiple lines of investigation to validate their major claims. However, there are a few – mostly minor – issues that should be addressed before publication.

Major issues:

1) The authors state in the abstract that they have shown that the effect of FAM46C “is not attributable to poly(A) RNA polymerase activity.” In fact they have not tested that, even if several of their observations do support that conclusion. The Discussion states that there is a catalytically inactive FAM46C mutant, and I think its use to determine the role of polymerase activity in centriole number is required as part of this study. It is not necessary to repeat every experiment with that mutant - I think it would be sufficient to show that FAM46C polymerase dead mutant prevents Plk4-driven rosette formation, and that it inhibits Plk4 autophos in vitro.

2) The final experiment (matrigel invasion) is certainly exciting, but there is no effort to tie the reduced matrigel invasion to Plk4. It seems that some experiment is needed here to validate the notion of invasiveness as a readout of the ability of FAM46C to inhibit Plk4.

Minor issues:

1) There are several issues with the in vitro kinase assays:

A) What are the major bands in the coomassie stained gel in Fig 4A?

B) The rationale for using the Plk4 1-390 fragment in Fig 4 should be better described. Is the rationale that this construct lacks binding sites for Cep85 and STIL, or is there some other reason?

C) It might be worth commenting on the differences between Fig 4A and Fig 4B. On the surface it appears that as much as 100x more protein is being used in Fig 4B compared to Fig 4A. Is that the case, and if so why is this necessary? In addition, there is no FAM46C phosphorylation in Fig 4 A, but a radiolabeled FAM46C band appears in Fig 4B. Is this due to the GST tag on FAM46C, the greater amount of protein, or promiscuous activity of the Plk4 1-390 fragment?

2) Fig 4D seems more appropriate as part of Fig 1 or 2 showing that the excess Ctn2 foci observed upon FAM46C depletion are likely centrioles generated via the Plk4 pathway. While the use Sas6 incorporation as a readout of Plk4 activity is relevant in Fig 4, the result seems more consistent with an exploration of the FAM46C-depletion phenotype. Toward this point, the pattern shown for FAM46C depletion in Figs 1 and 2 is not so clearly the classical rosette pattern seen for Plk4 overexpression. By itself this is a non-issue, particularly given the “rosette” seen for Plk4 overexpression in Fig 3A, but it is another reason that Fig 4D would seem more at home to me in Fig 1 or 2 – it establishes the “rosette” phenotype more than do the data generated with Centrin staining.

3) Given that Plk4-depletion produces cells with a single centriole in cells depleted of FAM, the possibility exists that the co-siRNA phenotype represents epistasis as opposed to showing a requirement for Plk4 in production of excess centrioles by FAM46C-depletion. This is likely of little consequence given the preponderance of data from other experiments, but is worth consideration, and possibly worth commenting on.

4) I don't think it's true that the centriole cycle “continues uninhibited” in HU-arrested cells. If that were the case, you would not see rosettes but rather an ever increasing number of diplosomes. This is a minor issue, but there is probably a better way to state this.

5) Particularly with the data in Fig S2 it is reasonable to conclude that FAM46C localizes to centrioles without further ultra structural analysis. However, it is possible that the centriole field will not accept the excess centrin-positive structures as centrioles without EM or other ultra structural or functional validation. I list this as a minor issue rather than a major issue, because this could amount to me having to walk to school up hill both ways in the snow... we would be asked – and are nearly always asked – to do EM to validate that excess centrin-containing foci are actually centrioles, even when we have analyzed additional markers.

COMMSBIO-19-0751A
Response to Reviewers' Comments

Overview of additional experiments performed in response to the Editor's summary of Reviewers' concerns.

Taking the Reviewers' feedback in sum, there were four essential issues to be addressed, as outlined by the Editor:

1. Potential confounding effects of FAM46C RNA polymerase activity on Plk4 expression

We performed additional experiments to directly test the effect of a catalytically-inactive mutant FAM46C D90/92A on Plk4 kinase activity, and on centriole duplication. The results, now shown in Fig. S6, show a comparable inhibition of Plk4 autophosphorylation and suppression of the Plk4-induced centriolar overduplication and rosette phenotype as seen with wildtype FAM46C. (Further details provided below in specific responses to Reviewer #1, point 3 and point 7; and Reviewer #3, Major Issue #1)

2. Localization of FAM46C to the centriole throughout the cell cycle

To further characterize the localization of FAM46C in relation to other centriolar proteins throughout the cell cycle, we synchronized HeLa cells and stained for mother and daughter centriole markers, as well as FAM46C and centrin, in S, G2, M and G1 phases. The resulting immunofluorescence images are now shown in Fig. S3b, and show consistent preferential localization of FAM46C to the mother centriole throughout the cell cycle. (Further details provided below in specific responses to Reviewer #1, point 1 and point 2; and Reviewer #2, point 1)

3. Structural integrity of supernumerary centrioles in response to FAM46C depletion

We performed additional experiments using EM to examine the structure of the supernumerary centrioles generated in response to FAM46C depletion. As now shown in Fig. S2e, these appeared to have normal centriolar architecture, indicating that the excess centrin-positive structures were indeed centrioles. (Further details provided below in specific response to Reviewer #3, Minor Issue #5)

4. Causal link between FAM46C and Plk4 in Matrigel invasion experiments

As recommended, we performed additional invasion assays to explore the relationship between FAM46C and Plk4 function. As now shown in Fig. 6h, when Plk4 kinase activity was inhibited by centrinone B, FAM46C was no longer able to stimulate cancer cell invasion. This provides evidence of Plk4 dependence, and functional antagonism. (Further details provided below in specific response to Reviewer #3, Major Issue #2)

In submitting these detailed responses in conjunction with our substantially revised paper, we would like to sincerely thank the Reviewers for their careful reading of the manuscript, their insight and constructive criticisms. We feel that the additional experiments performed in response to the Reviewers' feedback have considerably strengthened our manuscript.

Detailed responses to concerns of individual Reviewers.

Reviewer #1

1. As recommended by the Reviewer, we have performed additional experiments to further characterize the centriolar localization of FAM46C. As shown in revised Fig. 1C, as well as revised Fig. S3b-d and Fig. S4a, we used CEP120 (preferentially localized to the daughter centriole) and ODF2 (preferentially localized to the mother centriole) staining to show that FAM46C is preferentially localized to the mother centriole. At the super-resolution level, FAM46C is seen to localize very similarly to, though not exactly the same as, Plk4 in untreated U2OS cells. This is conveyed in the cartoon representing mother-daughter centrioles now included in Fig. 1c.

As also recommended by the Reviewer, we interrogated the localization of FAM46C throughout the cell cycle, by synchronizing HeLa cells using Aphidicolin block and release. FAM46C was apparent at centrioles throughout S, G2, M and G1 phases, with a localization that appeared consistent with sustained preferential staining of the mother centriole (Fig. S3).

These results are described in pgh 5 of the Results section, and also noted in pgh 3 of the revised Introduction.

2. There was no evidence of a cell cycle block/arrest in cells depleted of FAM46C (Fig. 2d), making centriole endoreduplication an unlikely explanation for the centriolar overduplication phenotype we observed. We also performed additional experiments that examine the localization of FAM46C in the scenario of Plk4-induced centriolar rosette formation. In T-Rex YFP-Plk4 U2OS cells treated with Tet to induce Plk4-driven centriole overduplication, we observed an intriguing spatial relationship between Plk4, which stained the nascent centrioles that are arranged like the petals of a flower, and FAM46C, which was concentrated at the center of these petals (Fig. S3d).

The observations on cell cycle progression are described in pgh 4 of the Results section and the description of the FAM46C localization in centriolar rosettes are described in pgh 5 of the Results section.

3. We thank the Reviewer for recommending we test the dependency of FAM46C's effect on its NTase activity. We used the catalytically inactive D90/92A FAM46C mutant to perform new in vitro kinase experiments, and showed that the autophosphorylation of Plk4 was inhibited by mutant FAM46C, as it was by wildtype FAM46C (Fig. S6a). Furthermore, the catalytically inactive mutant also suppressed Plk4 driven centriolar amplification (Fig. S6b). These experiments provide important new evidence that supports our central hypothesis that FAM46C directly inhibits Plk4 kinase activity.

The description of these important new experiments is included in pgh 10 of the Results section and referred to in pgh 3 of the Discussion section.

4. While we have previously used pS305 specific antibody to identify active Plk4 (Rosario et al., *Oncogene* 2015), it has not been possible for us to generate reproducible results with

respect to quantification of active vs. total Plk4 at each subcellular location in relation to the level of FAM46C; this may be related to the simultaneous inhibition of Plk4 kinase activity and increase in total Plk4 protein level that appears to ensue upon forced expression of FAM46C. In future experiments, we hope to determine whether FAM46C regulates activity of Plk4 in cell protrusions, which may correlate with its regulation of cell invasion.

In cells depleted of FAM46C, Plk4 retains its centriolar localization. Additional detailed experiments must be performed to describe the effect of FAM46C depletion on active vs. total Plk4 levels at the multiple centrioles that are observed in consequence.

5. In response to the Reviewer's query about what region of the FAM46C sequence is required for its interaction with Plk4, we used 3 deletion mutants to show that the Plk4 interaction is dependent on the N terminal region, and does not require the entire DUF/NTase domain (Fig. S7d).

This finding is described in pgh 11 of the Results section.

6. We thank the Reviewer for suggesting we improve the quality of the image for the Coomassie blue stained gel showing the FAM46A control. We believe that it is now possible to appreciate that the quantity of FAM46A is increasing gradually over the lanes going from left to right, and that there is if anything slightly more FAM46A protein added than in the corresponding FAM46C control lanes (see modified Fig. 4a).

7. The Reviewer's question about the effect of FAM46C on Plk4 protein level, and/or the level of other Plk4 regulators, is an important consideration.

Since we knew that through its RNA polymerase activity, FAM46C regulates the expression of multiple death and survival genes at the mRNA level in a multiple myeloma cell line, we tested the effect of FAM46C depletion on viability and proliferation of cancer cells, finding no difference to account for the altered centriole profile (Fig. 2d).

In addition, FAM46C increases the level of wildtype Plk4 protein (Fig. 4c), an effect we argue may be attributable to its suppression of Plk4 kinase activity (consistent with this, the effect on kinase dead Plk4 protein level was minor, potentially reflecting interaction between endogenous active Plk4 and exogenous kinase dead Plk4, as noted in the Results section). Importantly, the effect on Plk4 protein (increased) level would not account for the effect of FAM46C on centriole number (decreased).

Furthermore, depletion of FAM46C in tumors in vivo results in reduced Plk4 expression (Fig. 5e). Nevertheless, depletion of FAM46C promoted tumour growth (Fig. 5d, f), which would not be predicted if its effect was mediated through the observed alteration in Plk4 expression level.

These are admittedly somewhat circumstantial arguments, however. We feel that the addition of the experiments using catalytically inactive FAM46C that has no RNA polymerase activity, showing that this mutant FAM46C nevertheless suppresses Plk4 kinase activity and Plk4-driven centriole duplication, more directly answers the question about whether the effect

of FAM46C is mediated through altered gene expression. This point has been expanded in pgh 3 of the Discussion section.

Reviewer #2

1. Centrosomal staining for FAM46C, whether endogenous or exogenous, was consistent and ubiquitous. We thank the reviewer for pointing out that we had not made this explicit. A sentence has been added to the second pgh of the Results section to so indicate.

As noted in response to Question 1 from Reviewer #1, we performed additional experiments to assess the localization of FAM46C throughout the cell cycle. As now shown in Fig. S3, FAM46C was apparent at centrioles throughout S, G2, M and G1 phases in HeLa cells that had been synchronized using Aphidicolin block and release. These results are described in pgh 5 of the Results section, and also noted in pgh 3 of the revised Introduction.

2. We thank the Reviewer for this suggestion. While previous reports of analyses of proteins isolated from purified centrosomal fractions using proteomic approaches such as Mass spectrometry and BioID have not indicated the presence of a protein that is consistent with FAM46C, conditions that maintain its binding and/or integrity may not have been employed. The consistent and selective localization of FAM46C to the mother centriole that we now demonstrate through new experiments included in the revised manuscript (Fig. 1c, Fig. S3) provide evidence that FAM46C is indeed a protein that localizes to the centriole. These results are described in pgh 5 of the Results section.

3. This is an important point, and we have taken the Reviewer's recommendation to examine in more detail the localization of FAM46C in Plk4-induced centriolar rosettes. Consistent with the above evidence that indicates preferential localization of FAM46C to the mother centriole, FAM46C staining was seen predominantly at the centre of the rosette structures, rather than on the nascent daughters (Fig. S3c,d). This distinguished its staining pattern from that of Plk4, which is a very interesting finding that will be pursued in future experiments. These results are described in pgh 5 of the Results section.

4. The possibility that the decreased centriole replication induced by transient RFP-FAM46C transfection could be due to a non-specific G1 cell cycle arrest related merely to the process of transient transfection was considered. The effect of RFP-FAM46C transfection is compared to that of transient RFP transfection (Fig. 2e). Transient RFP transfection did not interfere with the ability of Tetracycline to induce centriole overduplication in U2OS T-Rex YFP-Plk4 cells (Fig. 3a, Fig. S4d). Furthermore, RFP-FAM46C had differential effects in RFPshRNA vs. FAM46CshRNA cells (Fig. S4b), implying that the impact was other than a non-specific effect on cell cycle progression. This point is now more explicitly made in pgh 6 and pgh 8 of the Results section.

5. In our initial description of Plk4 expression levels in primary colorectal cancer (Macmillan et al., 2001 and thesis published by Jennifer C. Macmillan, University of Toronto 2001) we showed the range of T/NM values and illustrated the necessity to log transform these ratios to permit statistical analysis. The rationale for transformation to the log scale (Fig. 6a,b) is

to more clearly display the range of T/NM values that are less than 1, and permit display and analysis as a “normal” function with expected symmetrical distribution around the point ‘0’. The original publications are referred to in reference 8 (Kazazian et al., 2015).

6. We thank the Reviewer for the suggestion to explore the status and effect of FAM46C in other cancer types. We have a separate full-length manuscript in preparation that will examine expression levels and mutations in patients with primary gastric adenocarcinoma who have undergone resection, with correlation to clinical outcome. In addition to the above analysis of tissues in our own clinically-annotated tumour bank, our analysis of data available from the TCGA database suggests that depletion of FAM46C in gastric cancer tumour tissue is associated with an adverse prognosis. In direct response to the Reviewer’s recommendation, we are employing gastric cancer cells lines to further study the effects of FAM46C on gastric cancer cell adhesion and invasion, as well as transcriptional regulation.

Reviewer #3

Major Issues:

1. We thank the Reviewer for suggesting that we directly examine the effects of the catalytically inactive FAM46C mutant on Plk4 kinase activity and function. We have followed through on this with new experiments that demonstrate that the D90/92A mutant suppresses Plk4 autophosphorylation to a similar degree as wildtype FAM46C, and that this catalytically inactive mutant also suppresses Plk4-driven centriole overduplication, both in terms of the characteristic centriolar phenotype and the numbers of centrioles per cell (Fig. S6, described in pgh 10 of the Results section). These results strengthen the evidence that FAM46C is inhibiting Plk4 kinase activity directly, rather than through transcriptional regulation based on its RNA polymerase function.

2. We agreed with the Reviewer that it would be helpful to show additional experiments that tie the effect of FAM46C in these invasion assays to cellular Plk4 activity, and have now performed them (Fig 6h). Depletion of FAM46C had the predicted stimulatory effect on invasion. Importantly, that stimulatory effect was lost when cells were treated with the Plk4 kinase inhibitor Centrinone B, showing a causal link between FAM46C and Plk4 with regard to regulation of cellular invasion. We thank the Reviewer for this recommendation. The new experiments are described in pgh 13 of the Results section, and also noted in pgh 3 of the Introduction.

Minor Issues:

1. In vitro kinase assays

a) The major bands seen in the Coomassie stained gel are IgGs – heavy chain at about 50 kDa and heavy plus light chain at about 75 kDa.

b) We used Plk4 1-390 because we were able to generate sufficient quantities of it in bacteria, and it retains the ability to auto-phosphorylate. We had difficulties with the yield of full length His-tagged Plk4 protein from bacteria. This is now explained in a sentence that has been added to pgh 9 of the Results section.

c) The Reviewer is correct that larger amounts of both Plk4 kinase domain+linker and FAM46C were used in this in vitro kinase assay depicted in Fig. 4b vs. Fig. 4a. This was quite a different system, since the proteins were derived from bacteria rather than mammalian cells, were His- or GST- tagged, respectively, and were subjected to rigorous purification. We intentionally generated large amounts of protein for this experiment, as can be appreciated from the gel. This distinction from the method used in Fig. 4a is now more clearly stated in pgh 9 of the Results section. As to the explanation for the apparent minor radiolabeling of the GST-FAM46C band, all three of the potential explanations offered by the Reviewer could be valid. The lack of phosphorylation of FAM46C by full length wildtype Plk4, as illustrated in Fig. 4a and Fig. S6a, was very consistent when the kinase assay was performed using mammalian cellular extracts.

2. We had intended the Sas6 result to show the downstream impact of altered Plk4 kinase activity, and therefore included it with the kinase assay results, but the Reviewer makes an excellent suggestion to move this result up, not only to confirm the nature of the FAM46C depletion phenotype, but also to better illustrate that FAM46C depletion results in an actual “rosette” centriolar overduplication pattern. This is now Fig. 2c, and the Figure Legends and text have been edited appropriately.
3. We agree with the Reviewer that epistasis is another potential explanation for the failure of FAM46C depletion to induce centriole overduplication in cells depleted of Plk4. However, we also agree with the Reviewer’s opinion that the more likely explanation is that the effect of FAM46C depletion depends on the presence of Plk4 activity. We have added a comment about the relative likelihood of these potential explanations to pgh 7 of the Results section.
4. With regard to HU-arrested cells and the status of the centriole cycle: We agree that the word “uninhibited” is not appropriate, and have accordingly reworded the sentence in pgh 8 of the Results section.
5. As recommended, we now include EM verification that the excess centrioles found in FAM46C-depleted cells are structurally normal (Fig. S2e). This finding is described in pgh 3 of the Results section.

Reviewers' comments:

Reviewer #1 (Remarks to the Author):

The authors addressed some of my concerns in the revised manuscript. However, this reviewer is not convinced with the main claim that FAM46C specifically regulates the activity of Plk4 at centrioles, by the following reasons. This reviewer feels that the authors should seriously consider to improve these points.

1) Related to the comment 1: based on the detailed distribution of FAM46C at centrioles (Fig. 1C), FAM46C does not colocalize with Plk4. This is not consistent with the claim because, for example, the known Plk4-binding protein such as Cep152 is reported to colocalize with Plk4 around the mother centriole. Moreover, even when overexpressing Plk4, FAM46C localizes onto the Plk4 ring only very partially, suggesting that FAM46C could not directly bind to Plk4 at centrioles.

2) Related to the comment 4: The authors did not address the important point that FAM46C affects the Plk4 activity or its endogenous expression level at centrioles. If FAM46C directly binds to and inhibits Plk4, one would assume that Plk4 expression level at centrioles becomes increased. This assumption is based on the fact that the amount of Plk4 is negatively regulated by its trans-autophosphorylation and SCF/bTrcp-mediated ubiquitination status. Moreover, the phosphorylation status at centrioles can be tested using phospho-specific Plk4 antibodies (Sillibourne et al (2010)MBoC, Nakamura et al. (2013, Nat Commun). These important points were not yet addressed in the revised manuscript.

3) Related to the comment 5: The authors tried to narrow down the binding region of FAM46C to Plk4, but this experiment is not sufficient for confirming a significance of the interaction between FAM46C and Plk4. The authors showed that the N-terminal region is responsible for binding of FAM46C to Plk4, but the region should be further narrowed down to perform the functional assay with this deletion mutant. The binding-deficient FAM46C is expected to lack its inhibitory function against the Plk4 activity. This kind of experiment has not yet been done in the revised manuscript.

Moreover, does this binding region of FAM46C make sense in light of the fact that FAM46C, but not A,B and D, binds to Plk4? Is the N-terminal region specific to FAM46C?

4) Related to the comment 6: in Fig.4A, the authors replaced the image for the Coomassie blue staining gel showing the amount of FAM46A added in the experiment. However, unfortunately, it seems at least to this reviewer that the experiment was not adequately performed because FAM46A and FAM46C proteins should be quantified and shown in the same gel for accurate comparison. In the current data, these proteins were subjected into the separate gels, which makes it difficult to properly judge their ability for suppressing the Plk4 kinase activity. Moreover, there is a clear tendency that as FAM46A was added more, the autophosphorylation signal of Plk4 was attenuated. In Fig.4B, FAM46C inhibits the Plk4 kinase activity in vitro, but this seems not to be dependent on the amount of FAM46C added. Again, if the authors intend to show the importance of the interaction between FAM46C and Plk4 in the regulation of the Plk4 kinase activity, it would be more informative to use the binding-deficient mutant of FAM46C in these experiments. Overall, this biochemical assay is not convincing to support the main claim.

Reviewer #2 (Remarks to the Author):

The authors have satisfactorily replied to my comments 1, 3, 4 and 5 in the revised version of their manuscript.

With regard to my comment 2, they have added superresolution immunofluorescence imaging data, which certainly add valuable information but do not completely substitute for additionally proving the

centrosomal localization of FAM46C by Western blotting at the biochemical level. Has immunoblotting of centrosomal extracts with an antibody to FAM46C been tried?

In response to my comment 6 the authors now state that a manuscript on FAM46C expression and mutational status in gastric cancer is in preparation. Nevertheless, data on the impact of mutant FAM46C as described in multiple myeloma would certainly have strengthened the current manuscript, although not absolutely required.

Reviewer #3 (Remarks to the Author):

This is a revision of a manuscript previously submitted by Kazazian et al. showing that FAM46C localizes to centrioles where it acts as a tumor suppressor by counteracting Plk4-driven centriole over production. The authors have largely done a nice job responding to critiques both major and minor. For my part (previous Reviewer #2), my concerns were nicely dealt with, including the addition of experiments with catalytically inactive FAM46C to rule out any requirement of the polymerase activity, the use of the Plk4 inhibitor centrinone to directly tie invasiveness of FAM46C-depleted cells to Plk4 inhibition, and a not-insignificant EM analysis of centrioles in FAM46C-depleted cells. The authors treated even my minor concerns very thoughtfully, with the result that what I felt was an already strong manuscript has been significantly improved.

It seems to me that the authors also did a nice job responding to concerns from other reviewers. That said, the response to point 2 from Reviewer #1 gives me some pause. Specifically, the authors argue that it is unlikely that cells have undergone "endo-reduplication" because no cell cycle block was noted. I take to be a response to a comment from Reviewer #1 about whether "centriole reduplication or multiple daughter centriole formation against a single mother" is responsible for the excess centrioles in FAM46C-depleted cells. I'm not convinced that either mechanism suggested by Reviewer #1 requires a cell cycle block (though many use cell cycle block to observe the phenomena), so it's not clear to me that the authors were addressing the point as intended. Regardless, I think the nice EM shown in Fig S2E nicely addresses this, as it looks like it actually demonstrates a mother centriole that initiated two daughter centrioles (though examination of an adjacent serial section may be required to verify that both "things" sticking off the centriole in the middle are daughters). Otherwise, an already strong manuscript has been improved thanks to a thoughtful response to critiques.

Response to Reviewers' Comments on Revised Manuscript***Overview of additional experiments performed in response to the Editor's summary of Reviewers' remaining concerns.***

Taking the additional feedback from the Reviewers in sum, there were two key residual issues to be addressed, as outlined by the Editor:

1. Does FAM46C truly localize to the centrosome, similarly to Plk4?

Reviewer #1 questioned whether the immunofluorescence images we have included are truly representative, and recommended an alternative method to characterize the cellular location of FAM46C. Along the same lines, Reviewer #2 suggested that we purify centrosomal extracts and use immunoblotting to probe for FAM46C.

Response: To isolate centrosomes, we performed sucrose gradient fractionation of cell lysates from exponentially growing U2OS cells. To characterize the subcellular location(s) of FAM46C and Plk4, we co-transfected the cells with FLAG-Plk4 and RFP-FAM46C prior to lysis and fractionation, then separated proteins in the individual fractions via SDS/PAGE, followed by immunoblot for FLAG, RFP and gamma tubulin. As now shown in Fig. S3a, FLAG-Plk4 and RFP-FAM46C were present in the same fraction in which gamma tubulin was enriched. This result demonstrates that FAM46C is located in the centrosome, similarly to Plk4, supporting the evidence gathered from immunofluorescence images.

2. Is the inhibition of Plk4 kinase activity by FAM46C specific?

Reviewer #1 expressed concern that the suppression of Plk4 autophosphorylation we had observed in the presence of FAM46C might be nonspecific.

Response: To address this, we performed *in vitro* kinase assay experiments that show the relative effects of FAM46A and FAM46C run on the same gel, as recommended. With loading of comparable amounts of protein in the FAM46A lanes, Plk4 autophosphorylation remains strong, whereas it is markedly inhibited in the presence of FAM46C (now Fig. S7a). The reviewer further suggested that the non-binding FAM46C C terminal fragment should be included as a control in autophosphorylation experiments. In response, we performed new *in vitro* kinase assays that show the C terminal AA 193-391 fragment had no apparent effect on Plk4 kinase activity (Fig. S8d). The results of these new biochemical experiments have considerably strengthened the evidence that FAM46C specifically binds to and suppresses the kinase activity of Plk4.

In submitting these detailed responses in conjunction with our further revisions, my co-authors and I would like to thank the three reviewers for their thorough and thoughtful consideration of our revised manuscript. In response to their comments and recommendations, we performed several additional experiments, and now submit a revised manuscript strengthened by the new data thereby acquired. Below is a point-by-point response to each of the reviewers' comments that details these additional revisions.

Detailed responses to residual concerns of individual Reviewers.

Reviewer #1

1. The reviewer cautions against overinterpretation of co-immunoprecipitation experiments and immunofluorescence data regarding the potential interaction of FAM46C with Plk4, and continues to recommend other techniques to probe the intracellular distribution of FAM46C. In recognition of this, we undertook new experiments to isolate the centrosomal fraction by centrifugation through a sucrose gradient. U2OS cells were co-transfected with RFP-FAM46C and FLAG-Plk4 prior to fractionation, and serial fractions probed for the presence of the two proteins. As now shown in Supplemental Figure 3a, RFP-FAM46C and FLAG-Plk4 were both concentrated in the centrosomal fraction, marked by strong gamma tubulin signal. This result is described in paragraph 2 of the Results section of the revised manuscript, and provides additional evidence that FAM46C localizes specifically to centrosomes. We are grateful to the reviewer for recommending this technique for characterizing centrosomal proteins.

2. (re: previous comment #4) The reviewer questions whether the activity of Plk4 located at the centrosome is modified by FAM46C. We have investigated this by querying the ability of Plk4 to alter centriolar number and morphology, under conditions of increased and decreased FAM46C, with results shown in Figure 3 a,b and Supplemental Figure 7c. Our examination of SAS6 as a read-out of Plk4 activity further supports the conclusion that FAM46C inhibits Plk4 function (Figure 2c). We concede that while we have provided good evidence that FAM46C modulates the downstream function of Plk4 in centriole duplication, and that Plk4 and FAM46C interact physically in yeast 2 hybrid and co-immunoprecipitation experiments (and in addition that a FAM46C fragment that doesn't bind to Plk4 also fails to inhibit its kinase activity, as described under our response to comment #3, below), that both are localized to the centrosome according to sucrose gradient fractionation, and that both are localized to the centrosome throughout the cell cycle as assessed by immunofluorescence, there remains a possibility that Plk4 activity is regulated by FAM46C without a direct physical interaction between the two at the centrosome. The autophosphorylation of Plk4 at the centrosome could likewise be modified by FAM46C in the absence of a direct physical interaction between the two at the centrosome. We have carefully analysed our manuscript to ensure that we have not made the claim that Plk4 and FAM46C are physically interacting at the centrosome, and have accordingly modified the language of statements in the Abstract, in the Results section (paragraph 7), and in the Discussion section (paragraph 1). In addition, in paragraph 3 of the Discussion section, we acknowledge that the precise physical relationship of FAM46C to Plk4 at the centriole remains to be elucidated.

3. (re: previous comment #5) We thank the reviewer for recommending that we investigate the ability of the C terminal fragment of FAM46C (AA 193-391), which does not bind to Plk4, to inhibit its kinase activity. In a new *in vitro* kinase assay shown in Supplemental Figure 8d (representative of 3 independent experiments), we show that this non-interactive FAM46C fragment is unable to inhibit Plk4 autophosphorylation. This lack of inhibition was observed even at high protein levels of FAM46C 193-391. This important new information is

now described at the end of paragraph 11 in the Results section, the final paragraph under the subheading ***"FAM46C inhibits Plk4 autophosphorylation and interacts with the Plk4 kinase domain"***.

The sequence similarity is much higher between hFAM46C and FAM46C in other eukaryotes, than between hFAM46C and human paralogs FAM46A, B, and D, as shown in the dendrogram in Supplemental Figure 1a. Interestingly, the N terminus sequences of hFAM46A, B and D are indeed more divergent from that of FAM46C, than are the respective C terminus sequences. The divergence is most marked in the first 75 amino acids. We are grateful to the reviewer for suggesting this comparison, which is now noted in paragraph 11 of the Results section.

4. (re: previous comment # 6) With respect to the *in vitro* kinase assays we have performed to assess the effect of FAM46C on Plk4 autophosphorylation, the reviewer would like to ensure that the amount of FAM46A protein used as a control is similar to the amount of FAM46C. To address this, we have performed a new series of experiments and we show the relative effects of FAM46A and FAM46C by running them on the same gel, as recommended. As now shown in Supplemental Figure 7a, even with loading of comparable amounts of protein in the FAM46A lanes, Plk4 autophosphorylation remains strong, whereas it is markedly inhibited in the presence of FAM46C. The reviewer also questions the dose-response relationship between FAM46C and the attenuation of Plk4 autophosphorylation. This can be appreciated in Supplemental Figure 7b. The reviewer has suggested that the non-binding FAM46C fragment would be a better control in these autophosphorylation experiments. Accordingly, as described above, we performed new experiments to test the effect of the C terminal fragment 193-391 on Plk4 autophosphorylation. As noted, this fragment had no apparent effect on Plk4 kinase activity (Supplemental Figure 8d). The results of all of these new *in vitro* kinase experiments are described in paragraphs 9, 10 and 11, respectively of the Results section. We thank the reviewer for suggesting these experiments, the results of which have considerably strengthened our evidence that FAM46C binds to and suppresses Plk4 kinase activity.

Reviewer #2

1. (re: previous comment # 2) The reviewer asks about methods other than immunofluorescence that could verify the centrosomal localization of FAM46C. To address this, we performed sucrose gradient fractionation of cell lysates from U2OS cells that had been co-transfected with FLAG-Plk4 and RFP-FAM46C. As now shown in Supplemental Figure 3a, FLAG-Plk4 and RFP-FAM46C were present in the same fraction in which gamma tubulin was enriched. Stripping and re-probing separately with anti-FLAG and anti-RFP confirmed the identity of the two proteins as Plk4 and FAM46C, respectively, as noted in the legend accompanying Supplemental Figure 3a. This new finding is described in paragraph 2 of the Results section of the revised manuscript.

2. (re: previous comment #6) We appreciate the reviewer's suggestion to study the effect of catalytically inactive FAM46C, a mutant that lacks nucleotidyl transferase activity, on Plk4 kinase activity and centriole number. Supplemental Figure 7 in the revised manuscript focuses

on our investigation of FAM46C D90/92A mutant that has been shown to lack RNA polymerase activity in multiple myeloma cells. Here we find that FAM46C D90/92A inhibits Plk4 autophosphorylation to a similar extent as wild-type FAM46C. Furthermore, the FAM46C D90/92A mutant also suppresses Plk4-driven centriolar overduplication, similar to wild-type FAM46C. These results indicate that nucleotidyl transferase activity is not required for FAM46C's suppressive effects on Plk4 function, and imply that the inhibition is direct, rather than indirect, through altered RNA polymerization. Paragraph 10 of the Results section (under subheading "***FAM46C inhibits Plk4 autophosphorylation and interacts with the Plk4 kinase domain***") has been revised to highlight these important observations.

Reviewer #3

We appreciate the reviewer's comments and further feedback. The reviewer's point regarding the limitations of our argument about the relationship between centriolar phenotypes and cell cycle progression is well taken, and we have adjusted the wording of paragraph 8 of the Results section accordingly. In addition, in paragraph 3 of the Results section, we specifically draw attention to the appearance of two daughter centrioles apparently arising from the same mother centriole in the EM image now shown as Supplemental Figure 3b. This image is representative of others that demonstrate the same phenomenon. We thank the reviewer for highlighting the significance of this finding.

REVIEWERS' COMMENTS:

Reviewer #1 (Remarks to the Author):

The authors have addressed most of my concerns. Although they now show convincing data for the interaction between FAM46C and Plk4, it remains unclear that the activity state of Plk4 is locally regulated by FAM46C at centrosomes. Most of the data shown in this study are based on what would happen in the cytoplasm or in vitro. This point can be experimentally feasible to address, but it should be at least toned down throughout the manuscript.